# Dynamics of Endophytic Fungal Communities Associated with Cultivated Medicinal Plants in Farmland Ecosystem

**DOI:** 10.3390/jof9121165

**Published:** 2023-12-04

**Authors:** Chao He, Deyao Meng, Wanyun Li, Xianen Li, Xueli He

**Affiliations:** 1Institute of Medicinal Plant Development, Chinese Academy of Medical Sciences & Peking Union Medical College, Beijing 100193, China; hc891215@126.com; 2College of Life Sciences, Hebei University, Baoding 071002, China; mengdeyao2022@126.com (D.M.); liwanyun0623@126.com (W.L.)

**Keywords:** endophytic fungal diversity, co-occurrence network, functional predication, next-generation sequencing, cultivated medicinal plants

## Abstract

Microorganisms are an important component of global biodiversity and play an important role in plant growth and development and the protection of host plants from various biotic and abiotic stresses. However, little is known about the identities and communities of endophytic fungi inhabiting cultivated medicinal plants in the farmland ecosystem. The diversity and community composition of the endophytic fungi of cultivated medicinal plants in different hosts, tissue niches, and seasonal effects in the farmland of Northern China were examined using the next-generation sequencing technique. In addition, the ecological functions of the endophytic fungal communities were investigated by combining the sequence classification information and fungal taxonomic function annotation. A total of 1025 operational taxonomic units (OTUs) of endophytic fungi were obtained at a 97% sequence similarity level; they were dominated by Dothideomycetes and Pleosporales. Host factors (species identities and tissue niches) and season had significant effects on the community composition of endophytic fungi, and endophytic fungi assembly was shaped more strongly by host than by season. In summer, endophytic fungal diversity was higher in the root than in the leaf, whereas opposite trends were observed in winter. Network analysis showed that network connectivity was more complex in the leaf than in the root, and the interspecific relationship between endophytic fungal OTUs in the network structure was mainly positive rather than negative. The functional predications of fungi revealed that the pathotrophic types of endophytic fungi decreased and the saprotrophic types increased from summer to winter in the root, while both pathotrophic and saprotrophic types of endophytic fungi increased in the leaf. This study improves our understanding of the community composition and ecological distribution of endophytic fungi inhabiting scattered niches in the farmland ecosystem. In addition, the study provides insight into the biodiversity assessment and management of cultivated medicinal plants.

## 1. Introduction 

Endophytic fungi, which are distributed in plants within a wide range of habitats, are a group of microorganisms that exist inside healthy plant tissues without causing obvious disease symptoms to the host plants [1,2,3]. Endophytic fungi are capable of promoting plant growth, modulating plant development, and protecting host plants from a wide variety of biotic and abiotic stresses [4,5]. Some studies reported that the composition and distribution of endophytic fungal communities vary with plant species and ecological environment [6,7,8,9]. For example, the operational taxonomic unit (OTU) richness and community structure of leaf endophytic fungi were found to be related to plant identity [10]. Previous studies have documented that certain fungal genera, including *Balansia*, *Balansiopsis*, *Atkinsonella*, and *Echinadothis*, predominantly inhabit warm-season grasses, whereas *Acremonium* primarily colonizes cool-season grasses [11]. Different orchid plants within a shared habitat exhibit distinct preferences and prioritize specific symbiotic fungi [12]. Furthermore, the distribution of endophytic fungi in desert plants showed obvious differentiation in root, stem, and leaf tissues [13]. Sun et al. [14] confirmed that the host and tissue effects were remarkable in endophyte community assembly, but the host effect was stronger than the tissue effect. Photita et al. [15] found that the diversity of endophytic fungi was mainly affected by the season, precipitation, and average temperature. Kumar and Prasher [16] found that the endophytic fungal species diversity of *Dillenia indica* was the lowest in the winter and highest in the rainy season. According to Martins et al. [17], the fungal diversity of olive trees was higher in the roots than aboveground, and it was higher in spring than in autumn. However, most recent research has focused on the changes in endophytic fungal communities during seasonal turnover in only one plant niche.

In addition to host and environmental factors, plant microbiome construction and balance were found to be positively influenced by microbial interactions [18,19]. The co-occurrence network could be applied to characterize potential microbe–microbe interactions across different habitats [20,21] and to probe the characteristics of keystone OTUs and determine their importance in the community [22]. Zuo et al. [13] analyzed the microbial co-occurrence network in the roots, stems, and leaves of desert plants, where it was found that the interspecific connectivity of the root fungal network was the highest, whereas Hypocreales, as a critical taxon, participated in connecting above ground and underground fungal networks. Xiong et al. [23] demonstrated that rare taxa were prominently in the fungal networks and ecosystem functions (such as soil enzyme activities and crop yield). These results enhance our comprehension regarding the process of plant mycobiome assembly as well as highlight the critical role of certain species in maintaining the stability of plant microbial communities. However, our understanding of the co-occurrence network of endophytic fungi in different tissues of cultivated medicinal plants in farmland with frequent agricultural activities is limited, and research is needed to provide new insights into the interaction of fungi in the aboveground and belowground environments.

The endophytic fungi possess a high degree of genetic and functional diversity and endow various types of resistance and rich evolutionary pathways on plants. They influence plant growth, development, reproduction, host structure, as well as biodiversity [24]. Zhang et al. [25] found that the rhizomes of *Ligusticum chuanxiong* contain a high proportion of plant pathogens and that these fungal groups may also possess other physiological functions that are still to be discovered. Most of the endophytes of the three varieties of tobacco seeds were defined as pathogens. When the seeds leave the plant, they are sometimes rotten or mildewed under certain conditions. This finding also confirmed that many plant pathogens can grow in the nutritional manner of endophytes but never cause disease symptoms in the host, and the functions change across the different growth periods of plants [26]. Referring to the potential functions of endophytic fungi, especially whether host and external factors affect the nutritional functions of fungi, it is helpful to comprehensively explore the diversity of endophytic fungi and to tap into their resources.

The Anguo Medicine Planting Site is a typical farmland ecosystem. Some medicinal plants developed a symbiotic relationship with a variety of microorganisms [27]. However, little is known about how host species, tissue niches, and environmental factors interactively drive the assembly of endophytic fungi. In the present study, the endophytic fungi of five cultivated medicinal plants (*Bupleurum chinense*, *Astragalus membranaceus*, *Salvia miltiorrhiza*, *Lonicera japonica*, and *Atractylodes lancea*) in the Anguo Medicine Planting Site were selected. Furthermore, the dynamics of endophytic fungal communities in a total of 240 samples under different plant species, tissue niches (root and leaf), and seasonal alternations (summer and winter) were investigated using next-generation sequencing. We aimed to reveal the following: (1) if and how diverse endophytic fungal communities exist in cultivated medicinal plants in farmland environments; (2) their composition and distribution among different host plants, tissue niches, and seasons if present; and (3) whether endophytic fungi interact and what their functional importance is in ecological processes and the local environment. We aimed to understand the host–microbe relationship in cultivated plants and provide a basis for biodiversity evaluation and management of the farmland. These objectives were achieved by determining the endophytic fungal community composition, interactive relationships, and ecological functions in medicinal plant cultivation.

## 2. Materials and Methods

### 2.1. Study Sites and Sampling

The sampling sites were located in the Anguo Medicine Planting Site (38°42′ N, 115°32′ E) in the Hebei Province, China, which exists in a typical temperate continental climate. The climate parameters during the sample collection period are presented in Appendix A. In June and November 2021, samples were collected from experimental fields where a combination of five medicinal plants (*Bupleurum chinense*, *Astragalus membranaceus*, *Salvia miltiorrhiza*, *Lonicera japonica*, and *Atractylodes macrocephala*) were cultivated in a randomized manner. The five target plants collected had been growing for 2 years. Root and leaf tissue samples were collected from the aforementioned plant species. Each species designated six replicated plots, and within each plot, two healthy individuals of the same species were randomly selected, ensuring that there was an approximate distance of 50 m between each plant individual. Fresh leaves were cut using sterile scissors from the branches of the plant at the half-crown level, and root samples were collected at a depth of 30 cm from the soil layer below each plant. All the plant materials were collected on the same day, immediately sealed in plastic bags, and transported to the laboratory in an insulated container. The six subsamples of each plant in each small plot were mixed into one sample in equal amounts. We acquired a total of 240 samples for subsequent analysis, encompassing leaf and root samples gathered during both the summer and winter sampling periods.

### 2.2. DNA Extraction, PCR, and Illumina Miseq Sequencing

All the plant samples (leaves and roots) were washed under tap water in order to thoroughly remove impurities and sediment. Then, they were surface-sterilized via a consecutive soak for 4 min in 75% ethanol and 2 min in 5% sodium hypochlorite, followed by being rinsed three times with sterile distilled water. The treated samples were wiped dry with sterilized filter paper and placed in a refrigerator at −80 °C. Subsequently, they were subjected to DNA extraction. The plant endophytic fungal genomic DNA was extracted using the FastDNA^®^ Spin Kit for Soil DNA Extraction Kit (MP Biomedicals, Santa Ana, CA, USA). The DNA concentration and purity were determined using a NanoDrop 2000 UV-Vis spectrophotometer (Thermo Scientific, Wilmington, CA, USA), and the DNA quality was measured via 1% agarose gel electrophoresis.

The ITS1 region of the fungi was amplified by using an ABI GeneAmp^®^9700 PCR thermocycler (ABI, Vernon, CA, United States) with a fungal universal primer ITS1F (5′-CTTGGTCATTTAGAGAGTAA-3′) [28] and ITS2R (5′-GCTGCGTTCTTCATCGATGC-3′) [29]. The amplification reactions were performed in triplicate mixtures of 20 µL, including 10 μL of 2 × Pro Taq, 0.8 μL of forward primer (5 μM), 0.8 μL of reverse primer (5 μM), 10 ng of template DNA, and ddH_2_O. The PCR condition consisted of an initial denaturation at 95 °C for 30 min, 27 cycles of denaturation at 95 °C for 30 s, annealing at 55 °C for 30 s, an extension at 72 °C for 45 s, and then a final extension at 72 °C for 10 min. The PCR products were purified using an AxyPrep DNA Gel Extraction Kit (Axygen Biosciences, Union City, CA, USA) and eluted using Tris-HCl, and mass determination and quantification were conducted via 2% agarose electrophoresis and using a QuantiFluor-ST™ (Promega, Madison, WI, USA), respectively. Finally, the library was constructed (2 × 300 bp) and sequenced using the Illumina MiSeq PE300 platform (Illumina, San Diego, CA, USA) (Majorbio Bio-Pharm Technology Co., Ltd., Shanghai, China). Raw sequence data were submitted to the NCBI Sequence Read Archive (SRA) project accession number PRJNA942110.

### 2.3. Bioinformatics Processing

Raw fastq files were demultiplexed, quality-filtered, and merged via Trimmomatic and fast length adjustment of short reads (FLASH; Johns Hopkins University, Baltimore, MD, USA) [30]. Sequences that were less than 50 bp long and had an average quality score below 20 or those containing ambiguous bases were removed. The remaining high-quality sequences were merged based on the overlapping sequences between read pairs. Sequences with mismatches in the primer region were removed prior to further processing. Non-chimeric sequences were dereplicated, and any singletons were removed. Afterward, the optimized sequences were grouped into OTUs using UPARSE 7.1, based on 99% similarity [31]. The species final taxonomic annotation was performed using the RDP classification algorithm and a 90% confidence threshold in the UNITE database (v.18.11.2018). The RDP then collated the functional gene database from GenBank (Release 7.3; http://fungene.cme.msu.edu/ accessed on 6 April 2022) and obtained species annotation data. In order to eliminate potential bias due to the depth of the sequence, each sample was subsampled to the minimum counts of all samples. The subsample feature was used in MOTHUR (v.1.30.1) to flatten all the samples according to the minimum sequence number and remove annotated mitochondrial and chloroplast sequences from all the samples. Subsequently, rarefaction curves were assembled. Given the rarefaction curves of all samples approached asymptotes, the current sequencing strategy was sufficient to detect endophytic fungal diversity (Appendix A). The sequence numbers of each sample were normalized with the minimum sum of sequences among all samples using the “phyloseq” package in R (version 1.43.0) [32]. Finally, the OTU abundance table was obtained for statistical analysis.

### 2.4. Statistical Analysis

The experimental data were assessed for normal distribution and homogeneity of variance using SPSS 26.0 software. Additionally, the statistical significance of the obtained results was determined using Duncan’s multiple range test (*p* < 0.05). The data for the OTU richness of fungi were consistent with the normal distribution and homogeneity of variance. The bioinformatics analysis of the endophytic fungal community was conducted using the Majorbio Cloud platform (https://cloud.majorbio.com accessed on 28 March 2022). The specifics of the aforementioned are as follows: the alpha diversity indices, consisting of the number of OTUs, Chao1 richness, Ace richness, Shannon index, and Shannon even index (a Shannon index-based measure of evenness), were calculated using MOTHUR v1.30.1 [33]. Then, the samples were compared using the Student *t*-test. The formulas were as follows:(1)$$(1):\;\hat{S}\mathrm{Chao}1=S_{obs}+\frac{n_{1}(n_{1}-1)}{2(n_{2}+1)},$$
(2)$$(2):\;HSh=-\sum_{i=1}^{S_{obs}}\frac{n_{i}}{N}\ln\frac{n_{i}}{N},$$
(3)$$(3):\;\hat{S}=\left\{\begin{array}{c} S_{abund}+\frac{S_{rare}}{C_{ACE}}+\frac{n_{1}}{C_{ACE}}{\hat{\gamma }}_{ACE}^{2},for{\hat{\gamma }}_{ACE}<0.80\\ S_{abund}+\frac{S_{ACE}}{C_{ACE}}+\frac{n_{1}}{C_{ACE}}{\tilde{\gamma }}_{ACE}^{2},for{\hat{\gamma }}_{ACE}\geq 0.80 \end{array}\right.$$

The niche breadth of endophytic fungi was calculated to assess the capacity of different ecological niches to accommodate the endophytic fungi according to the formula *Bi* = 1/(∑R*j* = 1 *P*2 *kj*). Non-metric multidimensional scaling (NMDS) was performed based on the Bray–Curtis similarity, and the results were plotted in order to demonstrate the differences in the endophytic fungal community. Meanwhile, the nonparametric analysis of similarities (ANOSIM) was used to examine the significance of endophytic fungal differences between the different plants and tissues based on 999 permutations. The PERMANOVA test was applied to evaluate the percentage of variations explained by the hosts and seasons. The shared number of OTUs among different hosts and seasons was visualized using a Venn diagram. This analysis was conducted using R (version 3.4.0; R Core Team, Vienna, Austria) with the OTU count and taxonomy tables. The analysis involved examining the relative abundance of the top 30 fungal genera across various plant species through the utilization of a bubble chart. To ensure consistency, the data were normalized using relative mean values. The LEfSe [34] was measured to detect the abundant taxa of endophytic fungi among the different plants (LDA score > 2, *p* < 0.05). The FUNGuild database (https://github.com/UMNFuN/FUNGuild accessed on 16 March 2022) was utilized for the annotation of endophytic fungi. The database categorized these fungi into three distinct nutritional mode groups, namely pathotroph, symbiotroph, and saprotroph. A percent stacked column chart was constructed using Origin v9.0.

A network is a collection of all the possible interacting units in a system consisting of edge-connected nodes. Network analysis was used to determine the symbiosis mode of the endophytic fungi, where the OTUs with the top 100 abundance were first selected for a Spearman correlation test. In addition, all possible fungal OTUs with an absolute correlation coefficient with *p* < 0.01 and (rho) > 0.5 were selected for analysis. The analysis and visualization of the network structure were accomplished using Cytoscape v3.5.1 [35]. The structural parameters of the networks, such as the network connectivity, were determined. The modular and highly interconnected node groups were calculated by using the MCODE application. A module is a closely related area of the network, with more internal relationships than external ones [36]. The potential keystone taxa were generally considered to be those that exhibit higher values [37]. The ecological function prediction of endophytic fungi was based on the FUNGuild database. The stacked bar chart of species composition and functional prediction percentage of endophytic fungal communities was visualized using Origin 2019b 32 Bit. The degree d*i* of node *i* is given by d*i* = ∑_j∈V_^a*i*j^ [38].

## 3. Results

### 3.1. The Structure and Distribution of Endophytic Fungal Communities

The fungal sequences of the ITS1 region (3,945,221 in all) were clustered into 1234 OTUs at a 99% sequence similarity level in all samples after removing low-quality reads, chimeras, and singletons. The identified fungal OTUs represented 10 phyla, 35 classes, 89 orders, 217 families, 410 genera, and 621 species. The rarefaction curves at the OTU level for all samples nearly plateaued, indicating that most microbial diversity in the sample could be reflected at this sequencing depth (Appendix A).

The dominant classes and orders were Dothideomycetes (28.65–52.15%) and Pleosporales (12.68–31.01%) (Figure 1a,b). At the class level, Glomeromycetes were distributed mainly in *S. miltiorrhiza* and *A. lancea*, Leotiomycetes were distributed mainly in *L. japonica*, and unclassified Ascomycota were distributed mainly in *S. miltiorrhiza* and *B. chinense*. Agaricomycetes were found mainly in *L. japonica* and *A. lancea* (Figure 1a). In addition, Sordariomycetes (29.51–50.42%) were mainly distributed in the roots, while Dothideomycetes (33.3–72.46%) were found in the leaves. Sordariomycetes in the root system (1.93–50.42%) were more abundant in the summer, while Dothideomycetes in the leaves (11.2–72.46%) were more abundant in the winter (Figure 1c). At the order level, Hypocreales (12.13–39.46%) were mainly distributed in the roots, and Capnodiales (3.28–45.88%) were in the leaves. Hypocreales (1.62–39.46%) were mainly distributed in summer, while Capnodiales (0.79–45.88%) were mainly distributed in winter (Figure 1d).

The bubble chart shows the top 30 genera of endophytic fungi in five medicinal plants (Figure 2). Unclassified Ascomycota was mainly distributed in *S. miltiorrhiza* and *B. chinense*, whereas Leotiomycetes was mainly distributed in *L. japonica*. Agaricomycetes was mainly distributed in *L. japonica*. In addition, *Erysiphe*, *Vararia*, and *Alternaria* were highly abundant in *L. japonica*. *Fusarium* mainly occurred in *A. membranaceus*, whereas *Alternaria* and unclassified Mycosphaerellaceae were present mainly in *A. lancea*. Unclassified Ascomycota, *Cercospora*, and *Alternaria* were highly abundant in *S. miltiorrhiza*. Finally, unclassified Ascomycota was higher in *B. chinense*.

### 3.2. Alpha Diversity of Endophytic Fungi

In summer, the OTU richness of endophytic fungi in the roots was above the leaf tissue, but the pattern was opposite in winter (Figure 3a and Figure 4a). In summer, the endophytic fungal OTU diversity in the root was higher, and the values of the Chao1 and Shannon diversity of the endophytic fungi in *L. japonica* were the largest than those of other plants (Figure 3b–d).

In winter, the endophytic fungal OTU diversity in the leaf was higher, and the alpha diversity of the endophytic fungi in *A. lancea* was also higher (Figure 4b–d). Overall, the OTU richness and diversity of endophytic fungi in winter were higher compared with summer.

If hosts and seasons were not considered, the violin chart of the alpha diversity of endophytic fungi in the roots and leaves showed that the Chao1 index, Ace index, and Sobs index of endophytic fungi were higher in the root than in the leaf (Figure 5a). Moreover, the endophytic fungi in the root possessed higher niche widths (Figure 5b).

Venn diagrams showed the numbers of specific and shared OTUs of endophytic fungi in different tissues, seasons, and plant species (Figure 6). A total of 439 (42.83% of the total OTUs) were specific endophytic fungi in regard to leaf, whereas 317 (30.93%) were specific endophytic fungi of the root, and 269 (20.24%) were shared between them (Figure 6a). In addition, 279 OTUs (27.22%) were shared between summer and winter, and the proportion of unique OTUs was the highest among endophytic fungi in winter (48.68%) (Figure 6b). Notably, the OTUs of *S. miltiorrhiza* (16.98%) and *A. lancea* (16.68%) shared a greater proportion of taxa than other plants (2.46%). Additionally, 73 OTUs (7.12%) appeared simultaneously in five medicinal plants (Figure 6c). Among these, the proportions of OTU1161 (21.71%), OTU1085 (8.98%), and OTU885 (7.06%) were higher (Figure 6d).

### 3.3. Host and Seasonal Variation Drive Endophytic Fungi

In summer, the endophytic fungal composition across different plant species was significantly different within niche habitats (leaf: R = 0.5793, *p* = 0.001; Root: R = 0.5778, *p* = 0.001) (Figure 7a,b); the same was true in winter (leaf: R = 0.8489, *p* = 0.001; Root: R = 0.5644, *p* = 0.001) (Figure 7c,d). The tissue niche also had significant impacts on them (summer: R = 0.4667, *p* = 0.001; Winter: R = 0.4006, *p* = 0.001) (Figure 7e,f). Moreover, from summer to winter, the differences in the fungal composition of tissue niches were quite significant (root: R = 0.5703, *p* = 0.001; Leaf: R = 0.4023, *p* = 0.001) (Figure 7g,h). After conducting a homogeneity of variance test on all experimental group data, the results obtained from using PERMANOVA analysis showed that the variation in endophytic fungal communities was mainly explained by plant species (14.76%, *p* = 0.001), tissue niche (5.57%, *p* = 0.001), and season (5.46%, *p* = 0.001) (Table 1).

Linear discriminant analysis effect size (LEfSe) analysis revealed the most specific biomarker taxa associated with different plant species. The LDA score of the fungi of different plants was determined. The Figure 8a shows species whose LDA score is greater than the set value (LDA score > 2, less strict set to 2, and more strict was set as 4), that is, biomarkers with statistical differences between groups. The LEfSe identified *Cercospora* as the biomarker taxa for *S. miltiorrhiza*, unidentified *Didymerllaceae* for *L. japonica*, unidentified *Ascomycota* for *B. chinense*, *Verticillium* for *A. membranaceus*, and unidentified *Mycophaeosphaeria* for *A. lancea*. In addition, fungi with significant differences in different hosts were screened (Figure 8a). The differences in endophytic fungal composition between the root and leaf in different seasons were tested and analyzed at the genus level. In the roots, there were significant differences (*p* < 0.05) in the abundance of 15 fungal genera between summer and winter. Unclassified Ascomycota, *Vararia* sp., *Apiotrichum* sp., *Chaetomium* sp., *Podospora* sp., *Coprinopsis* sp., *Acremonium* sp., and *Penicillium* sp. showed significantly increased abundance in winter. While *Gibberella* sp., *Alternaria* sp., *Fusarium* sp., *Cladosporium* sp., *Neocosmospora* sp., *Aureobasidium*sp., and *Tausonia* sp. were significantly enriched in summer (Figure 8b). In the leaf, 15 fungal genera were found to differ significantly between summer and winter (*p* < 0.05). *Aspergillus* sp., unclassified Ascomycota, *Cutaneotrichosron* sp., *Simplicillium* sp., *Sarocladium* sp., *Gibberella* sp., and *Candida* sp. were significantly enriched in summer. Moreover, unclassified Mycosphaerellaceae, *Cercospora* sp., *Colletotrichum* sp., *Epicoccum* sp., *Corynespora* sp., *Setophaeosphaeria* sp., *Ochroconis* sp., and *Hannaella* sp. were significantly enriched in winter (Figure 8c).

### 3.4. Host and Seasonal Selection Affect Co-Occurrence Networks Complexity of Endophytic Fungi

The results of network analysis revealed that there were deviations in the endophytic fungal network structure of different plant species, tissue niches, and seasons. The endophytic fungi of *S. miltiorrhiza* and *A. lancea* had high network connectivity. The clustering coefficient (0.719), network density (0.295), and network centralization (0.193) of *S. miltiorrhiza* were higher than those of other plants. The network heterogeneity (0.639) and the number of network nodes (96) and edges (593) of *A. lancea* were greater than those of other plants (Figure 9a–e; Table 2). The leaf habitat had the highest network connectivity, as exemplified by the largest numbers of nodes (90) and edges (718) (Figure 9g,h; Table 2). The density (0.179 and 0.055), clustering coefficient (0.597 and 0.422), and centralization (0.276 and 0.099) of the leaf fungal network were found to be significantly higher than those of the root (Table 2). Molecular complex detection (MCODE) analysis indicated certain important fungal aggregation taxa in the network (seven in the leaf and eight in the root). Furthermore, these fungal communities were more closely connected, forming more complex small structures that may play an important role in farmland ecosystems (Figure 10a–d).

Taking into account the mutual interaction for endophytic fungi, an integral co-occurrence network was constructed for the endophytic fungi of medicinal plants. The network had a node count of 85 and an edge count of 165, with lower values of a clustering coefficient (0.431) and network density (0.082). We also found a higher network diameter (12) and the shortest average path length (4.583) in this network (Figure 9f; Table 2). This showed that host selection affects co-occurrence network complexity and endophytic fungi with different plant species and tissue constitute a complex network structure. In addition, the network structure of endophytic fungal flora also changed between seasons. Additionally, the numbers of nodes (97) and edges (364), clustering coefficient (0.549), and average shortest path length (3.936) of the networks were slightly higher in the winter than in summer, but the network density (0.078), network heterogeneity (0.582), network centralization (0.101), and network diameter (10) were slightly lower than those in summer (Figure 9i,j; Table 2).

Notably, the correlation analysis showed that the interspecific relationship between endophytic fungal OTUs in the network structure was mainly positive rather than negative. According to the node connectivity coefficient (degree) of each fungal OTU, 10 dominant keystone OTUs were identified in the two tissues, five plant species, and two seasons (Appendix A). At the order level, Glomerales predominated in the root, and Capnodiales were dominant in the leaf. There were no overlapping keystone species in the fungal communities among the different tissue niches. Moreover, Hypocreales was dominant in summer, and Glomerales was dominant in winter. Capnodiales, Helotiales, Pleosporales, Venturiales, Hypocreales, Chaetothyriales, and unclassified Glomeromycota were the key groups identified in the endophytic fungal network of the five medicinal plants.

### 3.5. Functional Prediction of Endophytic Fungi

We conducted a functional characterization analysis on 1256 OTUs of fungi. Our findings indicate that 79.6% of the fungal OTUs were successfully characterized, while the remaining 20.4% of the fungal OTUs could not be characterized. Among the functional fungal OTUs that were successfully matched, they were categorized into three levels based on credibility assessment: “highly likely”, “likely”, and “possible”. The proportions of these categories in the result were 12%, 37.7%, and 29.9%, respectively. The functional prediction of fungal communities with five medicinal plants showed that three trophic mode groups could be classified according to different resource utilization methods, including pathotroph, symbiotroph, and saprotroph (Figure 11). The endophytic fungal communities of *B. chinense*, *S. miltiorrhiza*, *A. membranaceus*, and *L. japonica* were mainly pathotrophs and symbiotrophs, while those of *A. lancea* were mainly pathotroph and saprotroph (Figure 11a). We assigned fungal OTUs to the specific trophic groups and then further subdivided them into 23 specific ecological guilds. Among them, ectomycorrhizal showed a difference among the five hosts (*p* < 0.05). The undefined saprotroph was found to have a significant difference among the five hosts (*p* < 0.01). For *B. chinense*, plant pathogen (36.41%), undefined saprotroph (24.11%), and the root-associated biotroph (15.47%) were the primary ecological guilds. For *S. miltiorrhiza*, plant pathogen (28.33%), undefined saprotroph (18.78%), and endophyte (13.19%) were the primary ecological guilds. For *A. membranaceus*, plant pathogens (22.56%), endophytes (16.92%), and animal pathogens (15.16%) were the primary ecological guilds. For *A. lancea*, plant pathogen (24.84%), undefined saprotroph (18.12%), and root-associated biotroph (14.04%) were the primary ecological guilds. For *L. japonica*, plant pathogen (26.33%), undefined saprotroph (21.92%), and root-associated biotroph (12.88%) were the primary ecological guilds (Figure 11b).

From summer to winter, pathotroph in the root decreased, while saprotroph increased (Figure 11c). In addition, endophytic fungi in the root increased the three ecological guilds, which are the following: undefined parasite, algal parasite, and root-associated biotroph (Figure 11d). However, from summer to winter, both pathotroph and saprotroph in the leaf increased (Figure 11c), and the endophytic fungi in the leaf also increased the three ecological guilds, which are the following: animal parasite, algal parasite, and endomycorrhiza (Figure 11d).

## 4. Discussion

### 4.1. Taxonomic Characteristics of Endophytic Fungi

In the present study, 1256 OTUs of endophytic fungi belonging to 10 phyla were recovered from all the samples. In detail, Ascomycota were the most abundant, and a few were Basidiomycotina. The dominant classes and orders were Dothideomycetes (28.65–52.15%) and Pleosporales (12.68–31.01%). Ascomycota members are saprophytic, promoting nutrient recycling in farmland. Studies of endophytic fungi, based on culture-dependent molecular approaches, showed that *Alternaria* sp. and *Chaetomium* sp. were dominant fungi in *L. japonica* [39], *Alternaria alternata* and *Chaetomium* sp. were dominant fungi in *B. chinense* [40], and *Alternaria* sp. and *Aspergillus* sp. were dominant fungi in *S. miltiorrhiza* [41]. This finding suggests that notable variations exist in the dominant fungal species across various plant species. This finding aligns with Whitaker’s study, which indicated notable variations in endophytic fungi among host species and host populations within 18 species of Asteraceae at a specific site. However, no distinctions were observed between host subfamilies [42]. Here, Hypocreales were distributed mainly in summer and dominated the endophytic fungal network in summer. Studies on the endophytic fungal communities of mulberry (*Morus* spp.) in different seasons (spring/autumn) showed that the spring samples harbored higher Hypocreales taxa [43], which could preliminarily predict that the abundance of Hypocreales taxa was lower in the colder autumn and winter. The members of Dothideomycetes were mainly distributed above ground; similarly, previous studies have also shown that Dothideomycetes were the most dominant fungal group above ground [44]. In addition, Dothideomycetes were identified as saprophytic fungi that play a role in xylem and deciduous layer decomposition and material cycling [45]. Capnodiales were distributed and also dominated the endophytic fungal network in the leaf. Capnodiales belong to leaf surface saprophytes, which affect host photosynthesis because of shading. However, Capnodiales, as plant endophytes, generally do not cause disease in their hosts and even have positive effects, such as growth promotion and pest resistance [46]. OTU885 (*Cercospora* sp.), which had a higher proportion of fungi common to the five plants, was also a key species in the fungal network. Ascochyta blight has long been a cercospora leaf spot and has become more common in recent years [47]. The possible benefits of *Cercospora* sp. for the host plant remain to be analyzed.

### 4.2. Host and Seasonal Variation Influence Diversity of Endophytic Fungal Communities

The alpha diversity, niche breadth, and unique fungal OTUs of endophytic fungi in the roots were greater than those in the leaves. This may be due to microbial communities colonizing the roots and being transported to above ground. Previous studies have confirmed that microbial communities of plants are gradually filtered from non-rhizosphere soils, rhizoplane roots, the endosphere, the phylloplane, and the leaf endosphere [10]. Some studies have also shown that different microbial communities in aboveground and belowground plant tissues may originate from vertical transmission from seeds to seedlings [48]. However, in winter, the endophytic fungal richness and diversity of leaf tissue were higher than those of root tissue. As endophytic fungal species may reproduce in harsh environments (such as the leaf layer), fungal communities in aboveground tissues have a greater degree of coexistence due to stress tolerance [49]. Leaves can be colonized by airborne spores, whereas roots mainly colonize through inoculum in nearby soil. In winter, the rhizosphere environment conditions were more uniform and stable than those above ground, which may be one of the reasons for the lower species richness and diversity in the root. Researchers reported that various bacterial and fungal genera show significant preferences for different host types [50]. For example, *Ferrimicrobium* tends to colonize non-mycorrhizal plant tissues [51], while *Nigrospora* is more likely to appear on ectomycorrhizal plant leaves [52,53]. These confirm that host selection (i.e., plant species and tissue niche) dominates in shaping the plant microbiome assembly.

Compared with plants in summer, plants in winter have a richer endophytic fungal OTU. Previous studies of rice (*Oryza sativa*) have also yielded the same results [54]. Numerous studies have shown that seasonal variations in climate (e.g., temperature, precipitation) have a remarkable impact on the diversity of aboveground fungal communities. Furthermore, seasonal changes affect the change in the ecosystem environment, leading to differences in the photosynthetic rates of host plants and their nutrient acquisition in the soil, thereby affecting the composition of the fungal community structure. As such, broad-scale environmental controls (e.g., temperature and precipitation) are the driving forces of the fungal communities [55,56]. Meteorological data show that, compared with May and June, October and November have lower temperatures (0 to 6 °C), lower rainfall (6.1 to 12.2 mm), and higher humidity (67% to 79%). Plant residues are the main source of primary inocula for plant fungal infections. When the air is moist and precipitation is low, fungi preferentially germinate spores on plant residues [57].

### 4.3. Host and Seasonal Variations Drive the Distribution of Endophytic Fungal Communities

Studies have shown that the construction of the plant microbiome is primarily determined by tissue niches and host species. The endophytic microbiome assembly is shaped more strongly by the host than by seasonal factors, according to our results. Similarly, studies on desert plants have shown that the distribution of endophytic fungi in different host plants is characterized by high heterogeneity and dispersion [13]. Some previous studies on the root and aboveground zonal microbiomes of poplar and sugarcane revealed that different tissue niches (i.e., roots or leaves) contain different microbial communities [7,8]. The endophytic fungal communities of the root or leaf also differ significantly between summer and winter. For example, the abundance of the rhizosphere microorganisms is climate-dependent; it is also worth noting that Pseudomonadaceae has a negative-feedback effect on long-term precipitation [58]. Hernández-Tasco conducted an evaluation of the microbial diversity, behavior, and frequency of endophytic fungi in the leaves of *S. magnifica*, *S. schiffneri*, and *S. speciosa* over a span of three years. The study revealed that both the year and season exerted a substantial influence on the abundance of fungal genera in plant leaves. Furthermore, in comparison to species-related factors, the α and β diversity indices were found to be more strongly affected by the year [59]. The microbial community in this coniferous forest was also shown to vary seasonally, and plant photosynthesis appeared to be a major driver of seasonality [60]. Thus, we can conclude that plant microbial communities also demonstrate the capability to vary across seasons, possibly owing to the seasonality of precipitation and temperature.

LEfSe analysis and differential species analysis verified the occurrence patterns of the endophytic microbiome structure. Furthermore, it was also determined that they were different in different plants, tissue niches, and seasons, thereby indicating that endophytic fungi have certain host preferences and organ/tissue specificity [61]. This result was consistent with the assumptions that this may be a result of different species pools in different habitat regions, as well as their niche assignment, because of differences in microbial life histories. With the turnover of seasons, the endophytic fungal flora changed in the root and leaf. A total of 15 species were evidently different between summer and winter in both the root and leaf, but these fungi were not the different species that were among the five plant species. Similar studies confirmed this conclusion, whereby endophytic fungi, in great measure, exhibit an “only” tissue preference [62].

### 4.4. Host and Seasonal Selection Affect the Co-Occurrence Network Complexity of Endophytic Fungi

There are various ecological relationships between microorganisms, ranging from mutualism to competition, that, in addition to niche preferences, shape differences in microbial abundances. Recently, the network technique has been frequently used in microbial studies as an important means by which to identify keystone species that maintain network stability and complexity [63]. Species in the community interact with each other through relationships (such as predation, competition, and mutualism) and are interconnected to form a network. Co-occurrence networks can be used to study the reasons for the presence of certain microorganisms in the same tissue, as well as to statistically identify the highly connected taxa in the community and screen out key microbial groups in the community [64,65]. Research studies have employed this methodology to identify the co-occurrence patterns among microbial communities. For instance, Sun et al. [66] conducted a study examining the carbon and nitrogen cycling genes within soil microbial networks in Tibet and a control group. The researchers discovered notable disparities in the key genes (module hubs and connectors) between the grazing conditions and the control group. Hu et al. [67] discovered that the implementation of long-term mulching fosters synergistic relationships among species within bacterial and fungal symbiotic networks, thereby mitigating interspecies competition. The results of the co-occurrence network analysis demonstrated that the composition of the networks differed between the plant species and tissue niches.

In one study, the phylloplane and rhizoplane were found to act as a bridge for the interaction between the host and the environment [68,69]. The study showed that the aboveground endophytic fungal networks are more powerful and stable than the belowground farmland ecosystems. Different from desert habitats, the close fungal interactions in the root may be associated with strategies for desert plants to adapt to arid environments [13]. We speculated that the complexity of the belowground microbial network was relatively low despite the rich diversity of the root microbial community because of the high yield and volatility of the farmland ecosystem. We found that the endophytic fungal flora of medicinal plants was mainly positively correlated, which proved that the endophytic fungi in the farmland were more mutualistic with each other. However, we also found that the negative edges of the co-occurrence networks gradually increased from the aboveground to belowground niches, which meant that the endophytic fungi increased competition in the aboveground niche. When leaves are exposed to sunlight, photosynthesis may promote endophytic fungal interactions. It has also been proposed that competitive microbial interactions can have a positive impact on microbiome stability [70,71].

Different plants have their own unique microbial communities. Studies have found that there is a “core microbiota” under the functional redundancy of plant-related microbiota, which, as a subset of the plant microbiota, carries genes essential for host health [72]. The dominant taxa and biomarkers, as potentially keystone taxa, are important in terms of in-microbiome assembly and ecosystem functions [73]. In this study, the potential keystone taxa were preliminarily identified according to the node connectivity coefficient (degree) of the network. The network structure of the endophytic fungal community changed little between the seasons, but the keystone taxa in the network changed significantly. The Capnodiales, Helotiales, Pleosporales, Venturiales, Hypocreales, Chaetothyriales, and unclassified Glomeromycota formed the core fungal group in the endophytic fungal network of the medicinal plants. It is worth noting that Capnodiales are the dominant taxa in the leaves, but Glomerales do not have a high relative abundance in the roots. Similarly, Helotiales, Venturiales, Chaetothyriales, and unclassified Glomeromycota were very low in abundance in the plants. Studies have shown that this does not contradict the concept that dominant taxa affect the function of microbial communities because of their high abundance, while keystone taxa may be able to selectively alter other members. Therefore, regardless of its abundance, it can exert its impact [45]. Taken together, these results suggested that plant compartments may provide different niches for specific microbiota. They can recognize signaling molecules and adapt the immune system in each niche [74]. The LEfSe analysis showed that *Cercospora*, unidentified *Didymerllaceae*, unidentified *Ascomycota*, *Verticillium*, and unidentified *Mycophaeosphaeria* were the indicator taxa of *S. miltiorrhiza*, *L. japonica*, *B. chinense*, *A. membranaceus*, and *A. lancea*, respectively. Combined with the previous results, these taxa have a high abundance in individual plants, and they may play a key role in regulating the host adaptation, inhibition of pathogens, and plant tolerance to stress, such that the indicator biota of each plant is regarded as one of their “core microbiota”. As such, these endophytic fungi are regarded as potential keynote taxa of the Anguo Medicine Planting Site.

### 4.5. Host and Seasonal Variation Affect the Ecological Functions of Endophytic Fungi

The endophytic fungal community was mainly composed of pathotrophs and symbiotrophs. This indicates that endophytic fungi might first reduce the impact of plant immunity on themselves through pathological characteristics and then co-exist with host plants and exchange resources with hosts in order to obtain nutrients. However, the endophytic fungal community of *A. lancea* was mainly composed of pathotrophs and saprotrophs, which might obtain essential nutrient resources for the purposes of survival by degrading plant root and leaf epidermal cells with the help of saprophytic characteristics, as well as then promoting the growth of *A. lancea* through mutual symbiosis and interactions between microorganisms. The abundance of 23 ecological guilds varied among the endophytic fungi in the different plants, among which the ectomycorrhizal and undefined saprotroph differed the most among the five species, thereby indicating that the fungal communities performed different ecological functions in the different plants. In the ecological guilds, we found orchid mycorrhizal, endomycorrhizal, arbuscular mycorrhizal, root-associated biotroph, ectomycorrhizal, etc. This has potential application value in promoting the growth of medicinal plants and improving secondary metabolites [75,76].

In farmland ecosystems, seasonal change is one of the factors that contribute to the functional transformation of the endophytic microbiome of medicinal plant tissues. From summer to winter, the saprotroph of endophytic fungi increased in the root and leaf. A similar phenomenon was also found in the deciduous temperate forests; that is, the rot taxa of the saprotrophic reached their seasonal maximum on the freshly fallen litter in autumn [60]. In addition to the saprotrophs, the pathotrophs of the endophytic fungi in the leaf also increased, which is consistent with previous studies—that is, when the balance between the host and endophytic fungi is destroyed, endophytic fungi may become saprophytic or pathogenic [77]. This may be due to leaves becoming senescent in winter, whereby the plant defense system is broken down and the fungus is able to acquire nutrients more easily and thus colonize plant tissues more widely. In this way, endophytic fungi may manifest as saprophytes [78]. Alternatively, necrotrophic fungi become pathotrophs by killing the host and later obtaining nutrients from inside cells [79]. In conclusion, the host and season drive the functional transformation of endophytic fungal communities in medicinal plants, and through the selection of a microbial community structure and function, medicinal plants can become tenacious.

## 5. Conclusions

This study found that the endophytic fungal taxa and communities have host (species identities and tissue niches) and season preference patterns, and endophytic fungi assembly was shaped more strongly by host than by season, which indicated high heterogeneity for endophytic fungi associated with cultivated medicinal plants in farmland environment. The results showed that endophytic fungal communities were mainly composed of Dothideomycetes and Pleosporales. Network analysis showed that the fungal network connectivity was more complex in the leaf than in the root, and the interspecific relationship between endophytic fungal OTUs in the network structure was mainly positive rather than negative. The functional prediction of fungal communities associated with medicinal plants showed that there were three trophic groups, pathotroph, symbiotroph, and saprotroph. In addition, from summer to winter, pathotroph in the root decreased, and saprotroph increased, while both pathotroph and saprotroph in the leaf increased. This study adds to our understanding of endophytic fungal communities and distribution in cultivated medicinal plants and provides insight into the biodiversity assessment and management in the farmland.

## Figures and Tables

**Figure 1 jof-09-01165-f001:**
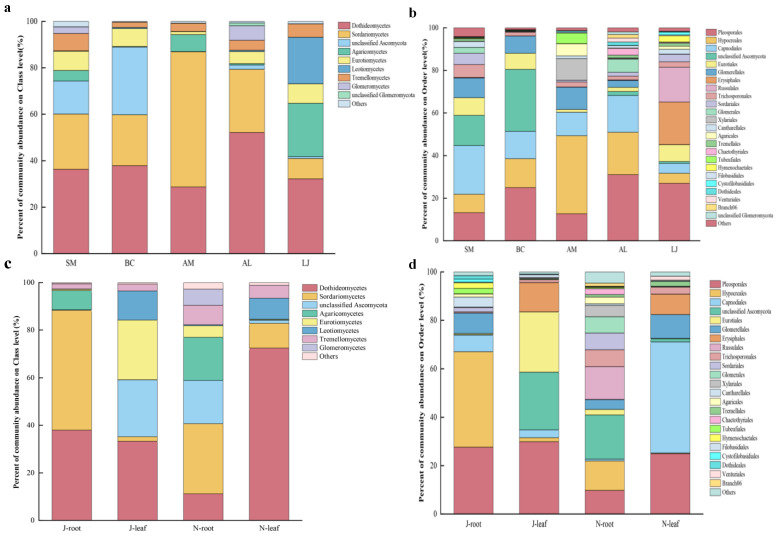
Relative abundance of the endophytic fungi at class and order levels in medicinal plants. (**a**) Endophytic fungi of plants at the class level. (**b**) Endophytic fungi of plants at the order level. (**c**) Endophytic fungi of root and leaf at the class level. (**d**) Endophytic fungi of root and leaf at the order level. Note: the fungal class and order represent < 0.01% of the total reads of endophytic fungi, which were all assigned to “Others”. Abbreviations—SM: *Salvia miltiorrhiza*; BC: *Bupleurum chinense*; AM: *Astragalus membranaceus*; AL: *Atractylodes lancea*; LJ: *Lonicera japonica*. J: June; and N: November.

**Figure 2 jof-09-01165-f002:**
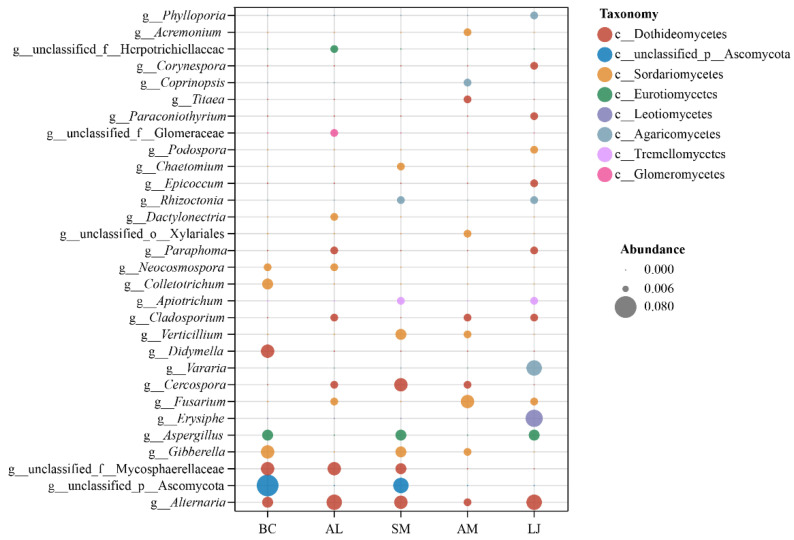
The bubble chart of the species annotation information, as well as the relative abundance of medicinal plants at the genus level. Note: the size of the circle indicates relative abundance, and the color of the circle shows the species annotation information at the class level. Abbreviations—LJ: *Lonicera japonica*; AM: *Astragalus membranaceus*; AL: *Atractylodes lancea*; SM: *Salvia miltiorrhiza*; BC: *Bupleurum chinense*.

**Figure 3 jof-09-01165-f003:**
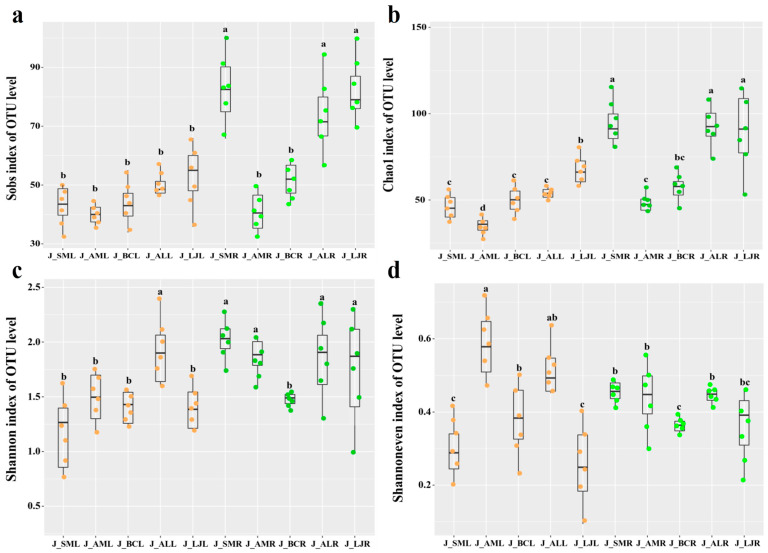
Operational taxonomic unit (OTU) richness and alpha diversity estimates of the fungal communities in June. (**a**) Sobs index at the OTU level. (**b**) Chao index at the OTU level. (**c**) Shannon index at the OTU level. (**d**) Shannon even index at the OTU level. Note: Different letters above the error bars indicate significant differences at *p* < 0.05. The orange dots represent the OTU of the leaf and the green dots represent the OTU of the root. Abbreviations—SM: *Salvia miltiorrhiza*; BC: *Bupleurum chinense*; AM: *Astragalus membranaceus*; AL: *Atractylodes lancea*; LJ: *Lonicera japonica*. R: Root; L: Leaf; J: June.

**Figure 4 jof-09-01165-f004:**
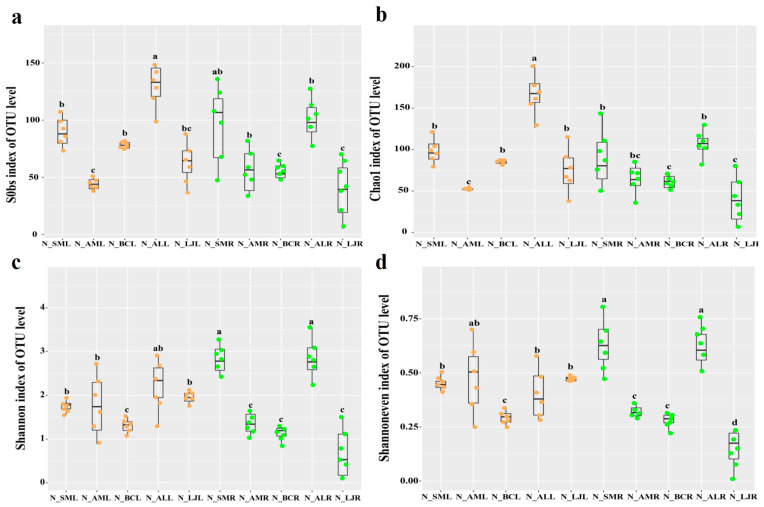
Operational taxonomic unit (OTU) richness and alpha diversity estimates of the fungal communities in November. (**a**) Sobs index at the OTU level. (**b**) Chao index at the OTU level. (**c**) Shannon index at the OTU level. (**d**) Shannon even index at the OTU level. Note: Different letters above the error bars indicate significant differences at *p* < 0.05. The orange dots represent the OTU of the leaf and the green dots represent the OTU of the root. Abbreviations—SM: *Salvia miltiorrhiza*; BC: *Bupleurum chinense*; AM: *Astragalus membranaceus*; AL: *Atractylodes lancea*; LJ: *Lonicera japonica*. R: Root; L: Leaf; N: November.

**Figure 5 jof-09-01165-f005:**
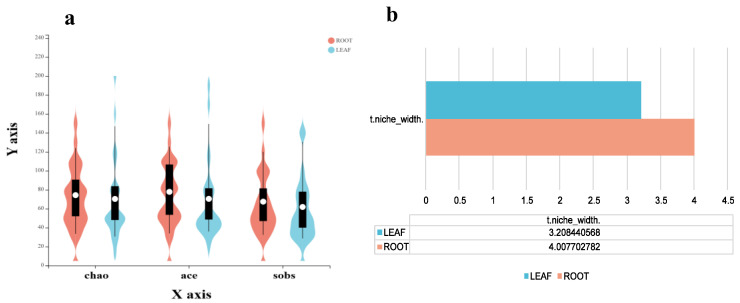
Diversity index and niche width of endophytic fungi in different tissues of plants. (**a**) Alpha diversity, (**b**) Niche width. The white circle in the figure represents the median.

**Figure 6 jof-09-01165-f006:**
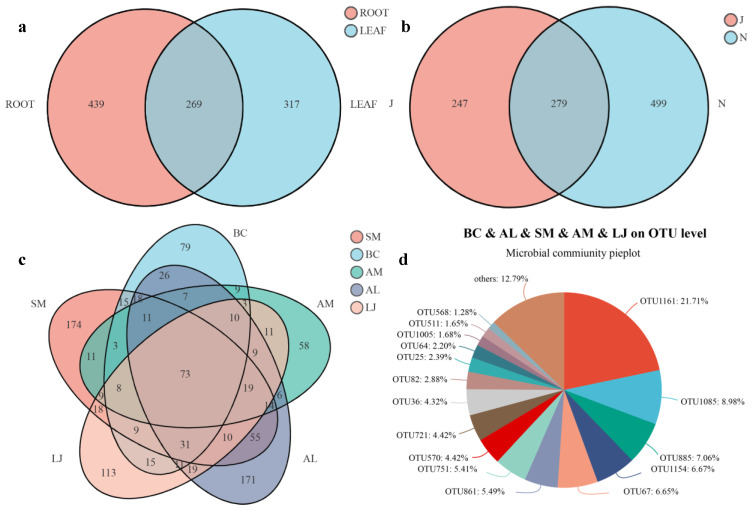
Venn diagram displaying the fungal composition among the different plant species, tissue niches, and seasons. (**a**), Tissue niches. (**b**), Seasons. (**c**), Plant species. (**d**), BC & AL & SM & AM & LJ on OTU level.Abbreviations—SM: *Salvia miltiorrhiza*; BC: *Bupleurum chinense*; AM: *Astragalus membranaceus*; AL: *Atractylodes lancea*; LJ: *Lonicera japonica*. J: June; and N: November.

**Figure 7 jof-09-01165-f007:**
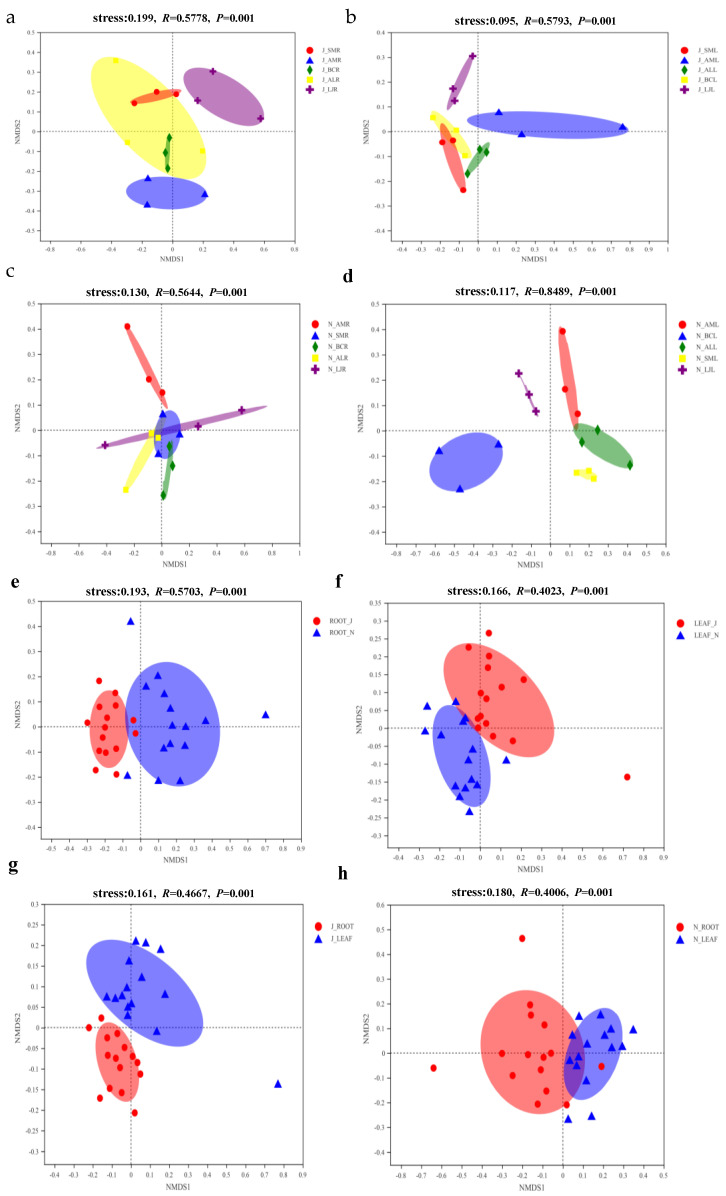
Nonmetric multidimensional scaling (NMDS) ordination of the fungal community composition among different plant species, tissues, and seasons. (**a**) Root in June. (**b**) Leaf in June. (**c**) Root in November. (**d**) Leaf in November. (**e**) Root in June and November. (**f**) Leaf in June and November. (**g**) Leaf and root in June. (**h**) Leaf and root in November. Note: ellipses in the plots denote 95% confidence intervals for the centroids of endophytic fungi among the different plant species, tissues, and seasons. Abbreviations—SM: *Salvia miltiorrhiza*; BC: *Bupleurum chinense*; AM: *Astragalus membranaceus*; AL: *Atractylodes lancea*; LJ: *Lonicera japonica*. R: Root; L: Leaf; J: June; and N: November.

**Figure 8 jof-09-01165-f008:**
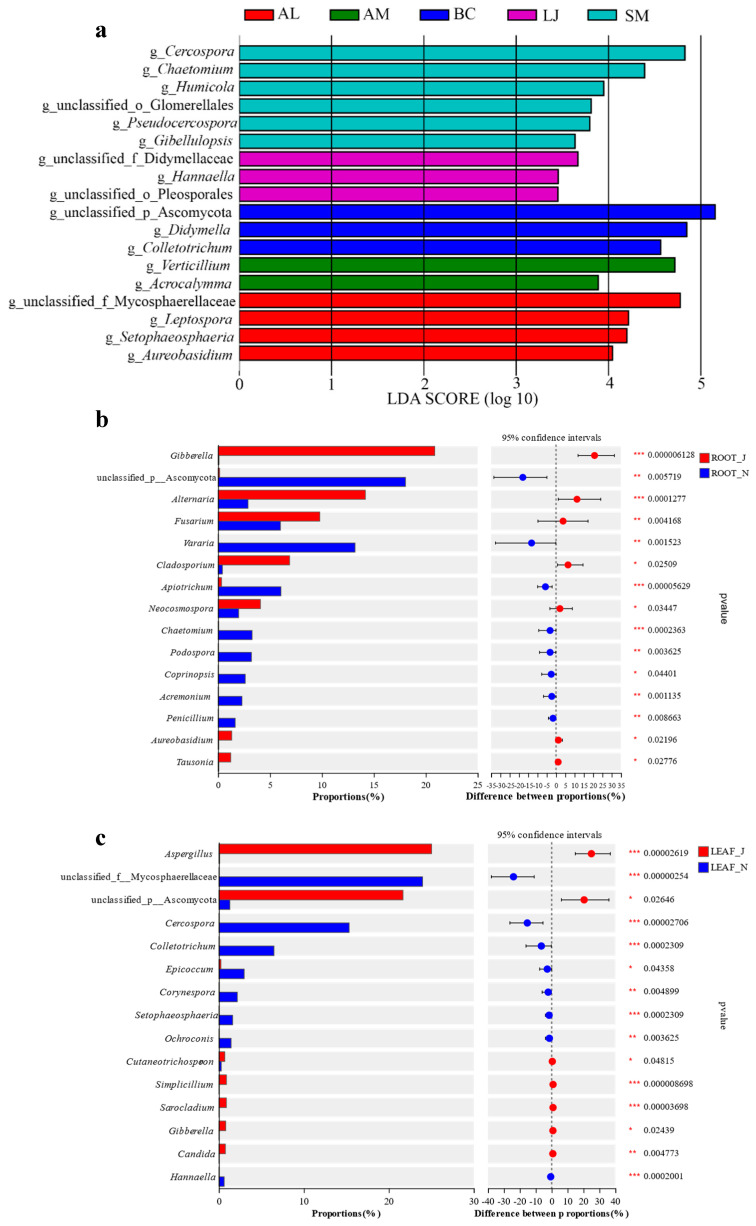
Differential abundance analysis of abundant fungi. (**a**) Indicator fungus with LDA scores of 2 or greater in medicinal plants. The LDA effect size (LEfSe) algorithm was used on the species level to determine the taxa that were differentially represented between the different host species. (**b**) Root in June and November. (**c**) Leaf in June and November. Note: asterisks indicate the significance levels, * *p* < 0.05, ** *p* < 0.01, and *** *p* < 0.001. Abbreviations—SM: *Salvia miltiorrhiza*; BC: *Bupleurum chinense*; AM: *Astragalus membranaceus*; AL: *Atractylodes lancea*; LJ: *Lonicera japonica*. J: June; and N: November.

**Figure 9 jof-09-01165-f009:**
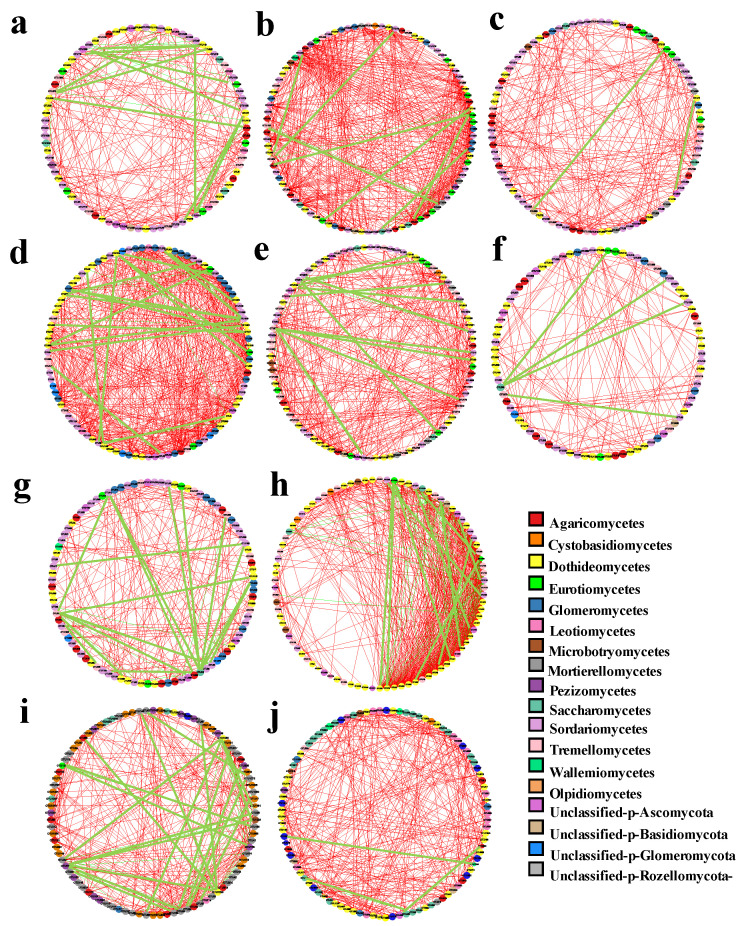
Co-occurrence networks of microbial taxa in the fungal communities. (**a**) BC; (**b**) SM; (**c**) AM; (**d**) AL; (**e**) LJ; (**f**) total endophytic fungi of plants; (**g**) root; (**h**) leaf; (**i**) June; (**j**) November. Note: The colors of the circles indicate different classes; the sizes of the circles indicate the relative abundance of the fungus. Edges represent significant interactive correlations between the pairs of OTUs. Red edges indicate positive relationships, and green edges indicate negative relationships. Abbreviations—SM: *Salvia miltiorrhiza*; BC: *Bupleurum chinense*; AM: *Astragalus membranaceus*; AL: *Atractylodes lancea*; LJ: *Lonicera japonica*.

**Figure 10 jof-09-01165-f010:**
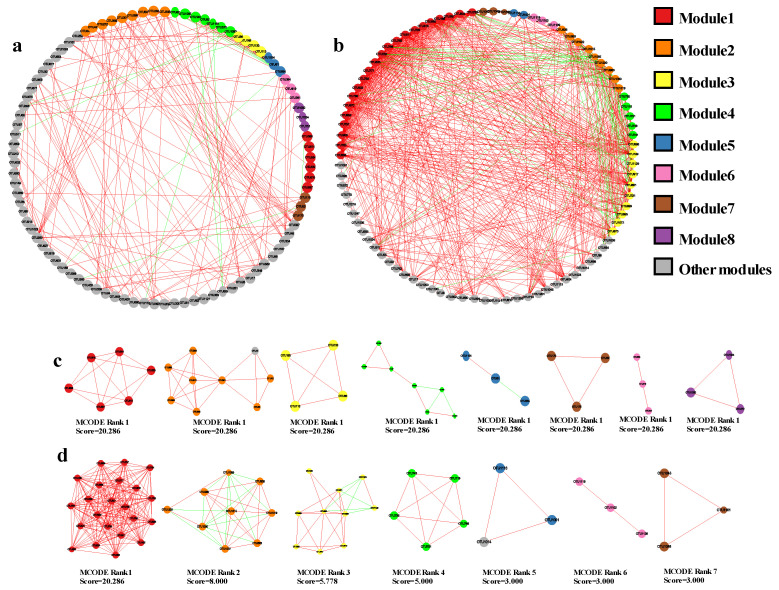
MCODE analysis showing the modular groups of highly interconnected nodes in networks. (**a**) Co-occurrence networks of root; (**b**) co-occurrence networks of leaf; (**c**) root MCODE module; (**d**) leaf MCODE module. Note: the colors of the circles represent the different modules, the sizes of the circles represent the relative abundance of the fungi, the red edges indicate positive relationships, and the green edges indicate negative relationships.

**Figure 11 jof-09-01165-f011:**
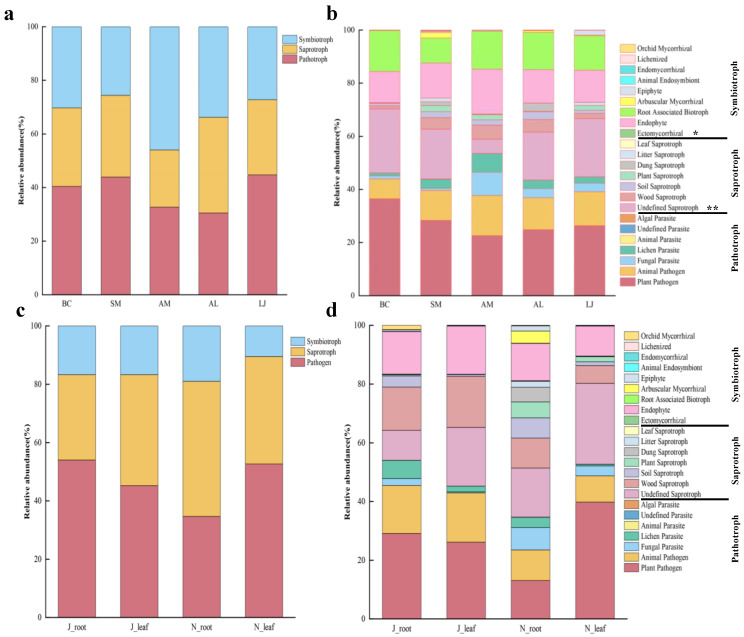
Functional prediction of endophytic fungal communities in different plant species and different tissue parts based on the FUNGuild database. (**a**,**b**) Plants species; (**c**,**d**) root and leaf. Note: The * symbol indicates the significant difference *p* < 0.05. The ** symbol indicates the significance level intervenes between *p* < 0.01. Abbreviations—BC: *Bupleurum chinense*; SM: *Salvia miltiorrhiza*; AM: *Astragalus membranaceus*; AL: *Atractylodes lancea*; LJ: *Lonicera japonica*. J: June; and N: November.

**Table 1 jof-09-01165-t001:** Effects of the plant species, tissue, and season on the fungal community structure, based on PERMANOVA.

Variables	ALL Samples	
	R^2^ (%)	Pr (>F)
Plant species	14.76	0.001
Tissue niche	5.57	0.001
Season	5.46	0.001
Plant species * Tissue niche	30.60	0.001
Plant species * Season	28.90	0.001
Tissue niche * Season	16.17	0.001
Plant species * Tissue niche * Season	62.22	0.001

Note: “*” represents the interaction relationship between different influencing factors.

**Table 2 jof-09-01165-t002:** Structural attributes of the networks of endophytic fungi for the different plant species, tissues, and seasons.

	Leaf	Root	BC	SM	AM	AL	LJ	June	November	Total
Nodes	90	93	84	95	91	96	95	92	97	85
Edges	718	235	243	554	263	593	390	345	364	165
Clustering coefficient	0.597	0.422	0.653	0.719	0.690	0.671	0.638	0.511	0.549	0.431
Network density	0.179	0.055	0.070	0.295	0.100	0.141	0.110	0.094	0.078	0.082
Network heterogeneity	0.742	0.598	0.500	0.631	0.428	0.639	0.617	0.722	0.582	0.560
Network centralization	0.276	0.099	0.089	0.193	0.071	0.148	0.131	0.233	0.101	0.110
Network diameter	8	10	11	9	13	10	9	12	10	12
Average shortest path length	2.670	4.318	4.979	3.168	5.572	3.959	3.584	3.819	3.936	4.583

## Data Availability

Data will be made available on request. Raw sequence data were submitted to the NCBI Sequence Read Archive (SRA) project accession number PRJNA942110.

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
