# Peer review of "Dynamics of Endophytic Fungal Communities Associated with Cultivated Medicinal Plants in Farmland Ecosystem"

_jof, 2023, doi:10.3390/jof9121165_

Round 1
Reviewer 1 Report
Comments and Suggestions for Authors
This is an interesting paper describing medicinal plants' fungi diversity and community composition; the authors try to infer how the community interacts in and with the plants to unveil the fungi's ecological roles. This is an interesting and well-written paper; the subject is relevant, considering the little information about microbial communities in plants of no agricultural importance. I have just some suggestions on how this paper could be improved.
Please review the tables and figures numbers; they must be consecutive, differentiating from those for supplementary material.
Please provide information about the study site, such as the age of the plants or the culture, climate conditions (temperature, relative moisture), and if it is a monoculture for each species.
Please describe more clearly why there are 120 samples: Do they correspond to five species, six plots, two tissues (leaf and root), and two stations?
And what are the subsamples integrated by? Are the six plots repetitions? Were composed samples used? This is unclear.
Table S1: The first column does not permit the identification of which orders correspond to each segment.
Which UNITE and Mothur database versions were used? Which FUNGuild version was used? Please report
Which FUNGuild categories were considered to integrate the "endophytic" fungi rating? Please specify.
P7, L229: Authors say, "Sordariomycetes (1.93-50.42%) preferred to occur in summer". Be careful because there is a clear difference between root and leaf: in summer, Sordariomycetes are scarce in leaf.
P9, L245: Figure 2 does not show Agaricomycetes in A. lancea but in A. membranaceus.
Please correct "OUT" to "OTU" in Figures 3 and 4 legends.
Figure 9 legend. Please review the text; it is quite confusing. Are the node colors correspond to the order or the classes of fungi? There is repetitive text.
The figure resolution is deficient, and many details are not clear; I couldn't distinguish the green edges.
P18, L386-388: I do not find any connection between the text and Table S2 cited.
Figure 11. The figures' legends are not visible, and the reading is incomprehensible. Please complete the figure description, specifying what is represented in each panel.
P19, L434: "Hypocreales" is mentioned twice in the same sentence; please remove one of them.
P20, L442-443: Please correct "Capnodiales were distributed mainly in the leaf, and also dominated the endophytic fungal network in the leaf" to "Capnodiales were distributed and also dominated the endophytic fungal network in the leaf"
I would have liked the authors to discuss the 73 OTUs sharing the five plant species concerning the community interactions, their role in the plants, and their ecological roles. Are these results consistent with the network analysis?
Author Response
Thank you very much for your valuable comments and advice. In the revision, we carefully considered the editor’s and reviewers’ comments and suggestions. The actions taken in the revision and answers and explanations in response to the queries are listed below.
Responses to the reviewers’ comments:
Reviewer 1
Comment 1:Please review the tables and figures numbers; they must be consecutive, differentiating from those for supplementary material.
Response: Thanks for your comment. We have checked and organized the numbering of all the tables and figures.
Comment 2:Please provide information about the study site, such as the age of the plants or the culture, climate conditions (temperature, relative moisture), and if it is a monoculture for each species.
Response: Thanks for your comment. We have revised. (Please see Line118-125 in the revised manuscript).
Comment 3:Please describe more clearly why there are 120 samples: Do they correspond to five species, six plots, two tissues (leaf and root), and two stations?And what are the subsamples integrated by? Are the six plots repetitions? Were composed samples used? This is unclear.
Response: Thanks for your comment and suggestions. We have made supplementary details. (Please see Line116-131 in the revised manuscript).
Comment 4:Table S1: The first column does not permit the identification of which orders correspond to each segment.
Response: Thanks for your comment and suggestions. There was a slight problem with the typesetting before. We have revised. (Please see Table S1 and line 132 in the revised manuscript).
Comment 5:Which UNITE and Mothur database versions were used? Which FUNGuild version was used? Please report
Response: Thanks for your comment. We have revised. (Please see Line167,171and 206 in the revised manuscript).
Comment 6:Which FUNGuild categories were considered to integrate the "endophytic" fungi rating? Please specify.
Response: Thanks for your comment. We have revised. (Please see Line 206-208 in the revised manuscript).
Comment 7:P7, L229: Authors say, "Sordariomycetes (1.93-50.42%) preferred to occur in summer". Be careful because there is a clear difference between root and leaf: in summer, Sordariomycetes are scarce in leaf.
Response: Thanks for your comment. We have revised. (Please see Line 242-244 in the revised manuscript).
Comment 8:P9, L245: Figure 2 does not show Agaricomycetes in A. lancea but in A. membranaceus.
Response: Thanks for your comment and suggestions. Based on other reviewers' comments, we have reanalyzed Figure 2 at the genus level. (Please see Fig.2 and Line 256-263 in the revised manuscript).
Comment 9:Please correct "OUT" to "OTU" in Figures 3 and 4 legends.
Response: Thanks for your comment. We have revised. (Please see Line 277-278,287-288 in the revised manuscript).
Comment 10:Figure 9 legend. Please review the text; it is quite confusing. Are the node colors correspond to the order or the classes of fungi? There is repetitive text.
Response: We have already deleted the duplicate parts according to your comment. (Please see Line 394-400 in the revised manuscript).
Comment 11:The figure resolution is deficient, and many details are not clear; I couldn't distinguish the green edges.
Response: Thanks for your comment. We have revised. (Please see Fig. 9 in the revised manuscript).
Comment 12:P18, L386-388: I do not find any connection between the text and Table S2 cited.
Response: Thanks for your comment and suggestions. We have revised. (Please see Table S2 and line417-420 in the revised manuscript).
Comment 13:Figure 11. The figures' legends are not visible, and the reading is incomprehensible. Please complete the figure description, specifying what is represented in each panel.
Response: Thanks for your comment. We have revised. (Please see Line 450-453 in the revised manuscript).
Comment 14:P19, L434: "Hypocreales" is mentioned twice in the same sentence; please remove one of them.
Response: Done. (Please see Line 467-468 in the revised manuscript).
Comment 15:P20, L442-443: Please correct "Capnodiales were distributed mainly in the leaf, and also dominated the endophytic fungal network in the leaf" to "Capnodiales were distributed and also dominated the endophytic fungal network in the leaf".
Response: Thanks for your comment and suggestions. We have revised. (Please see Line 475-476in the revised manuscript).
Comment 16:I would have liked the authors to discuss the 73 OTUs sharing the five plant species concerning the community interactions, their role in the plants, and their ecological roles. Are these results consistent with the network analysis?
Response: We sincerely thank the reviewer for this suggestion. We have modified it. (Please see Line 479-483 in the revised manuscript).
Reviewer 2 Report
Comments and Suggestions for Authors
Dear Authors,
I think your paper is not suitable for publication in its current form. To improve the paper, the following issues are to be addressed.
Major issues.
1., I think the amount of data collected does not warrant network analysis in my opinion, as if I understand correctly, data for a species are only made of a 2 x 2 x 3 set (season x tissue x replicates) - I think this is not enough for a statistically sound evaluation of co-occurrence. Perhaps building networks from the winter and summer data separately, as well as building networks from root and leaf data separately might provide data on the strength of the species factor. Please do a literature search for papers who evaluated the usefulness of the microbiome network analysis approach as a function of sample size. Also, a false discovery rate has to be calculated and your discussions limited to statistically significant co-occurences - this is not mentioned in the paper.
2., Though network studies are indeed widely used in microbiome studies, I think a sound discussion is missing. Listing lots of network characteristics does not add to the paper unless a biological interpretation is added. How do you translate network heterogeneity, centralization, etc. to something that is biologically meaningful?
3., I think that a 70% confidence in UNITE mathcing is possibly insufficient for giving single species resolution form ITS1 data only. Please add more details on how this threshold was chosen, as it can generate a relatively high false annotation rate.
4., Introduction: are the cited effects qualitative (different taxa appear in various samples, i.e. lack of co-occurrence) or quantitative only (only the proportions of the same set of taxa are different)? This is especially relevant among different species from the same environment. Please add this information to the first paragraph of the intro for the articles. Also include these (and possibly other similar papers to your discussion).
5., Discussion 4.1.: Compare your results to culture-independent studies on the species tested in the study or plants from the same plant subfamilies. A group found that foliar fungal endophyte varied significantly among host species, as well as host tribes, but not among host subfamilies in 18 Asteraceae species in a single site (Whitaker et al., 2020).
6., In 4.2: Your plants belong to various plant families, 4 of them are major plant families. Compare your findings to that in (Toju et al., 2019) and other similar papers (single site, many plant families).
7., In 4.3: Also note that the effects of year can supersede interspecies variances, as shown in (Hernández-Tasco et al., 2023). Perhaps the strength of the effect of season varies with species and/or habitat.
8., Figures are of poor quality. Higher resolution versions have to be provided (e.g. it is hard to read the color legend of Fig. 1., and it is not possible to decipher the color legend of Fig. 11.)
Technical/minor issues.
Table S1 is in the main manuscript. Table formatting does not enable the reader to tell where the main categories "root", "leaf" etc. begin or end.
Fig.2. Size legend is flawed. Perhaps add more resolution.
Use a compact letter display to show statistical differences. Fig 3. b and all subplots of Fig. 4 are especially looking odd.
Add abbreviation meanings to the captions of Fig. 3-4.
Best regards.
References
Hernández-Tasco, A.J., Tronchini, R.A., Apaza-Castillo, G.A., Hosaka, G.K., Quiñones, N.R., Goulart, M.C., Fantinatti-Garboggini, F., Salvador, M.J., 2023. Diversity of bacterial and fungal endophytic communities presents in the leaf blades of Sinningia magnifica, Sinningia schiffneri and Sinningia speciosa from different cladus of Gesneriaceae family: A comparative analysis in three consecutive years. Microbiological Research 271. https://doi.org/10.1016/j.micres.2023.127365
Toju, H., Kurokawa, H., Kenta, T., 2019. Factors Influencing Leaf- and Root-Associated Communities of Bacteria and Fungi Across 33 Plant Orders in a Grassland. Frontiers in Microbiology 10.
Whitaker, B.K., Christian, N., Chai, Q., Clay, K., 2020. Foliar fungal endophyte community structure is independent of phylogenetic relatedness in an Asteraceae common garden. Ecology and Evolution 10, 13895–13912. https://doi.org/10.1002/ece3.6983
Author Response
Thank you very much for your valuable comments and advice. In the revision, we carefully considered the editor’s and reviewers’ comments and suggestions. The actions taken in the revision and answers and explanations in response to the queries are listed below.
Responses to the reviewers’ comments:
Reviewer 2
Comment 1: I think the amount of data collected does not warrant network analysis in my opinion, as if I understand correctly, data for a species are only made of a 2 x 2 x 3 set (season x tissue x replicates) - I think this is not enough for a statistically sound evaluation of co-occurrence. Perhaps building networks from the winter and summer data separately, as well as building networks from root and leaf data separately might provide data on the strength of the species factor. Please do a literature search for papers who evaluated the usefulness of the microbiome network analysis approach as a function of sample size. Also, a false discovery rate has to be calculated and your discussions limited to statistically significant co-occurences - this is not mentioned in the paper.
Response: Thanks for your comment. We have provided detailed supplements regarding the details of the sampling. (Please see Line 116-131 in the revised manuscript).
Comment 2:Though network studies are indeed widely used in microbiome studies, I think a sound discussion is missing. Listing lots of network characteristics does not add to the paper unless a biological interpretation is added.How do you translate network heterogeneity, centralization, etc. to something that is biologically meaningful?
Response: Thanks for your comment. We have revised. (Please see line 554-560 in the revised manuscript).
Comment 3:I think that a 70% confidence in UNITE mathcing is possibly insufficient for giving single species resolution form ITS1 data only. Please add more details on how this threshold was chosen, as it can generate a relatively high false annotation rate.
Response: Thanks for your comment and suggestions. We have made supplementary additions to the relevant content. (Please see Line 165-171 in the revised manuscript).
Comment 4:Introduction:are the cited effects qualitative (different taxa appear in various samples, i.e. lack of co-occurrence) or quantitative only (only the proportions of the same set of taxa are different)? This is especially relevant among different species from the same environment. Please add this information to the first paragraph of the intro for the articles. Also include these (and possibly other similar papers to your discussion).
Response: We have revised according to your opinion. (Please see line 55-59 in the revised manuscript).
Comment 5: Discussion 4.1.: Compare your results to culture-independent studies on the species tested in the study or plants from the same plant subfamilies. A group found that foliar fungal endophyte varied significantly among host species, as well as host tribes, but not among host subfamilies in 18 Asteraceae species in a single site (Whitaker et al., 2020).
Response: Thanks for your comment and suggestions. We have revised. (Please see line 463-467 in the revised manuscript).
Comment 6:In 4.2: Your plants belong to various plant families, 4 of them are major plant families. Compare your findings to that in (Toju et al., 2019) and other similar papers (single site, many plant families).
Response: Thanks for your comment. We have revised.(Please see Line 498-501 in the revised manuscript).
Comment 7:In 4.3: Also note that the effects of year can supersede interspecies variances, as shown in (Hernández-Tasco et al., 2023). Perhaps the strength of the effect of season varies with species and/or habitat.
Response: Thanks for your comment. We have revised. (Please see Line 526-531 in the revised manuscript).
Comment 8:Figures are of poor quality. Higher resolution versions have to be provided (e.g. it is hard to read the color legend of Fig. 1., and it is not possible to decipher the color legend of Fig. 11.)
Response: Thanks for your comment and suggestions. We have revised. (Please see Fig.1 and Fig. 11 in the revised manuscript).
Comment 9:Table S1 is in the main manuscript. Table formatting does not enable the reader to tell where the main categories "root", "leaf" etc. begin or end.
Response: Thanks for your comment. We have revised. (Please see Table S2 in the revised manuscript).
Comment 10:Fig.2. Size legend is flawed. Perhaps add more resolution.
Use a compact letter display to show statistical differences. Fig 3. b and all subplots of Fig. 4 are especially looking odd.
Response: We have revised according to your suggestions. (Please see Fig.3 and Fig. 4 in the revised manuscript).
Reviewer 3 Report
Comments and Suggestions for Authors
Dear Authors,
please consider the following comments and opinions:
Mat & Met
Line 114: please rewrite the sampling part as it is not easy understandable if you have used 6 plants in total per species or there were 6 small plots with x plant individuals per small plot….
I mean try to follow: i) the number of plots in total, ii) the number of subplots per plot (if existed); iii) the number of plant individuals per subplot (if existed) or plot); iv) the number of samples per organ type and plant individual; v) the number of combined samples per organ and plant individual (required if case) and/or per organ and subplot (if case).
I mean, it is necessary to understand how your study design was planned so that to be reproducible and also to give information on how you considered your replicates.
Line 122: “the ten keystone species” -> you present genera and order mostly, please refer to what you have. Also, you need to explain the used formula/calculus for the degree of correlation (also, add the correspondent reference). Abundance needs to be explained also, with showing the formula or explanatory text (nr of total samples = biological samples or filtered seq and so on…)
Line 127: please explain again or for the first time “what do you consider a DNA sample”: did you combined multiple organ samples from the same plant/from different plants in the same plot and so. If you haven’t explained thoroughly in the “plot design” (up to name that “this workflow leads us to a DNA sample” then this is the place to do it. Anyway, it is best to recall “what was for you a dna sample” - subsamples = x, total samples of dna per subplot/plot etc..and how do you consider your replicates..if you have them further combined or not up to the dna sample.
Line 148: by far we can understand that you have “3 replicates of pcr products per dna sample” and that you sequenced them as such, it is correct? We need to understand if further you combined these sequences into a single file for a dna sample, and if you counted the same seq or eliminate it as presence/absence, or if you considered them as replicates for the stats. Please accurately explain your study design.
Line 154: please detail the algorithms used for the de-multiplexing steps, because if they are not open source/available then we cannot understand what criteria and parameters levels did you consider. Also, please details the ranges of parameters for the quality filtering
Line 159-160: please revise the first phrase as I for instance, do not understand this step. Please explain it simply and even step by step and explaining the reasoning better might help.
Line 169: if you did not violate the assumptions for anova, then why didn’t you use the parametric tests analysis instead? I mean a common way when you have normal distribution, and homogeneity of variance would be: anova -> post hoc multiple comparisons…
Instead you mention you have the assumptions well but you do not use the parametric workflow..just wondering…
Line 174: please cite in text which formula belongs to which index; also please correctly use their names: for “Shannon index” (also called Shannon entropy; for the formula: H, H’ or HSh = ….), for “Shannon’s evenness” (also called Pielou’s J; for the formula: J =…), Chao1 (estimator of total species richness; ÅœChao1 =…), ACE (abundance-based coverage estimator; for the formula: Åœ )
Line 188: Venn diagrams performed with…software/online platform please mention
Line 188: please take care as you haven’t identified at species level; therefore, use the appropriate taxonomic level (OTU = different sequences after the employed conditions for considering it unique; genus, order etc.) each time you address a feature/result etc. Please revise the entire manuscript.
Results
Line 213: I hope you are aware that a 97% threshold cannot offer you more than genus level; for the species level (ITS considering), you should be around 99% or even 99.6% to be suitable to distinguish among species. Although UNITE has a 98.5% threshold for hypothetically different seq species, you should apply for the in-house seq and the database the acknowledged thresholds for the fungal taxa.
(ref: https://doi.org/10.1093/femsec/fiw045; https://doi.org/10.1016/j.simyco.2018.05.001.
Figure 1: please take care at the y axis legend for the plot “c” as it is a class not an order fungal taxa graph
Figure 2: despite the fact that it would be wrong to have identified at species level using the threshold mentioned; it would be clearer to have the bubble graph per genera as many of them are identified at that level. Of course, this would imply to remake the analysis, as multiple species would gather inside same genus…
Figure 3 and Figure 4: legend -> please take care at “OUT” versus “OTU" + if you name your graph as “boxplots representing operational……” then there is no need to explain the median, intervals and so on : ) + there is no “ ***” for the p value and if you have used the brackets and the red dots to show which data groups are different, try using letters, as in the “a, b” plots it is not intelligible which one is different from which….
Figure 8 : idem opinion like for figure 2
Comments on the Quality of English Language
Please carefully revise the manuscript for minor English editing.
Author Response
Thank you very much for your valuable comments and advice. In the revision, we carefully considered the editor’s and reviewers’ comments and suggestions. The actions taken in the revision and answers and explanations in response to the queries are listed below.
Responses to the reviewers’ comments:
Comment 1: Line 114: please rewrite the sampling part as it is not easy understandable if you have used 6 plants in total per species or there were 6 small plots with x plant individuals per small plot….I mean try to follow: i) the number of plots in total, ii) the number of subplots per plot (if existed); iii) the number of plant individuals per subplot (if existed) or plot); iv) the number of samples per organ type and plant individual; v) the number of combined samples per organ and plant individual (required if case) and/or per organ and subplot (if case).I mean, it is necessary to understand how your study design was planned so that to be reproducible and also to give information on how you considered your replicates.
Response: We have rewrite according to your comment. (Please see Line 116-131 in the revised manuscript).
Comment 2:Line 122: “the ten keystone species” -> you present genera and order mostly, please refer to what you have. Also, you need to explain the used formula/calculus for the degree of correlation (also, add the correspondent reference). Abundance needs to be explained also, with showing the formula or explanatory text (nr of total samples = biological samples or filtered seq and so on…)
Response : Thank you very much for your valuable comments and advice. We added formulas and explanations. (Please see Line 419-420 in the revised manuscript).
Comment 3:Line 127: please explain again or for the first time “what do you consider a DNA sample”: did you combined multiple organ samples from the same plant/from different plants in the same plot and so. If you haven’t explained thoroughly in the “plot design” (up to name that “this workflow leads us to a DNA sample” then this is the place to do it. Anyway, it is best to recall “what was for you a dna sample” - subsamples = x, total samples of dna per subplot/plot etc..and how do you consider your replicates..if you have them further combined or not up to the dna sample.
Response: Thanks for your comment. We have added detailed sampling details in the "plot design" section. (Please see Line 116-131 in the revised manuscript).
Comment 4:Line 148: by far we can understand that you have “3 replicates of pcr products per dna sample” and that you sequenced them as such, it is correct? We need to understand if further you combined these sequences into a single file for a dna sample, and if you counted the same seq or eliminate it as presence/absence, or if you considered them as replicates for the stats. Please accurately explain your study design.
Response: Thanks for your comment and suggestions. In this study, we conducted the same collection operations in both summer and winter. In the two different seasons of sampling, a total of 48 samples were collected for each plant species, including 24 replicates of leaf and root samples for each plant species. Therefore, the cumulative total of DNA samples for these five different plant species is 240. We performed PCR treatment and high-throughput sequencing for each DNA sample.
Comment 5:Line 154: please detail the algorithms used for the de-multiplexing steps, because if they are not open source/available then we cannot understand what criteria and parameters levels did you consider. Also, please details the ranges of parameters for the quality filtering
Response: Thanks for your comment. We have revised. (Please see Line 158-164 in the revised manuscript).
Comment 6:Line 159-160: please revise the first phrase as I for instance, do not understand this step. Please explain it simply and even step by step and explaining the reasoning better might help.
Response: Thanks for your comment. We have revised. (Please see Line 169-171 in the revised manuscript).
Comment 7:Line 169: if you did not violate the assumptions for anova, then why didn’t you use the parametric tests analysis instead? I mean a common way when you have normal distribution, and homogeneity of variance would be: anova -> post hoc multiple comparisons…
Instead you mention you have the assumptions well but you do not use the parametric workflow..just wondering…
Response: Thanks for your comment. We have revised. (Please see Line 180-182 in the revised manuscript).
Comment 8:Line 174: please cite in text which formula belongs to which index; also please correctly use their names: for “Shannon index” (also called Shannon entropy; for the formula: H, H’ or HSh = ….), for “Shannon’s evenness” (also called Pielou’s J; for the formula: J =…), Chao1 (estimator of total species richness; ÅœChao1 =…), ACE (abundance-based coverage estimator; for the formula: Åœ )
Response: Thanks for your comment. We have revised. (Please see Line 190-192 in the revised manuscript).
Comment 9:Line 188: Venn diagrams performed with…software/online platform please mention
Response: Thanks for your comment and suggestions. We have revised. (Please see 200-202 in the revised manuscript).
Comment 10:Line 188: please take care as you haven’t identified at species level; therefore, use the appropriate taxonomic level (OTU = different sequences after the employed conditions for considering it unique; genus, order etc.) each time you address a feature/result etc. Please revise the entire manuscript.
Response: Thanks for your comment. We have revised. (Please see Line 202-204,256 in the revised manuscript).
Comment 11:Line 213: I hope you are aware that a 97% threshold cannot offer you more than genus level; for the species level (ITS considering), you should be around 99% or even 99.6% to be suitable to distinguish among species. Although UNITE has a 98.5% threshold for hypothetically different seq species, you should apply for the in-house seq and the database the acknowledged thresholds for the fungal taxa.
(ref: https://doi.org/10.1093/femsec/fiw045; https://doi.org/10.1016/j.simyco.2018.05.001.
Response: Thanks for your comment. We have revised. (Please see Line 225-228 in the revised manuscript).
Comment 12:Figure 1: please take care at the y axis legend for the plot “c” as it is a class not an order fungal taxa graph
Response: Thanks for your suggestion. We have modified it. (Please see Figure 1 in the revised manuscript).
Comment 13:Figure 2: despite the fact that it would be wrong to have identified at species level using the threshold mentioned; it would be clearer to have the bubble graph per genera as many of them are identified at that level. Of course, this would imply to remake the analysis, as multiple species would gather inside same genus…
Response : Thanks for your comment. We have revised. (Please see Fig.2 and Line 256-263 in the revised manuscript).
Comment 14:Figure 3 and Figure 4: legend -> please take care at “OUT” versus “OTU" + if you name your graph as “boxplots representing operational……” then there is no need to explain the median, intervals and so on : ) + there is no “ ***” for the p value and if you have used the brackets and the red dots to show which data groups are different, try using letters, as in the “a, b” plots it is not intelligible which one is different from which….
Response: Thanks for your comment and suggestions. We have revised. (Please see Fig.3 ,Fig. 4 and line 277-279,287-289 in the revised manuscript).
Comment 15:Figure 8 : idem opinion like for figure 2
Response: Thanks for your comment. We have revised. (Please see Fig.8 and Line 336-354 in the revised manuscript).
Reviewer 4 Report
Comments and Suggestions for Authors
Dynamics of endophytic fungal communities associated with cultivated medicinal plants in farmland ecosystem
The manuscript is well written and concise overall. The manuscript is scientifically sound for the most part but at least one clarification is needed.
Clarification needed: The authors use FUNGuild for functional evaluation; however, the authors need to state how many of the OTUs were functionally characterized and how many were not characterized. This is important because the authors discuss increased pathogens or increased saprotrophs, however it is important for the reader to know if this is based on 100% of the OTUs assigned to a guild or only 50% of the OTUs assigned to a guild. Typically, with NGS data a lot of the OTUs are unassigned so this needs to be added.
Line items:
Line 110: This is not the correct table. As per the manuscript, Table S1 contains the keystone species and Table 2 shows the climate parameters. Please adjust.
Line 120: This is awkward as written: suggestion: In total, 120 samples were processed for analyses.
Current table S1 (Keystone species). Unclassified Glomeromycota is not an Order so the column heading should be adjusted. This appears in two separate locations. Also, it is unclear where the total starts so a line separating the individual data vs the totals would visually make it clearer.
Line 140: The ITS2R primer is identical to the primer developed by White et al. 1990 so the authors should cite the original designer of the primer.
White, T.J.; Bruns, T.; Lee, S.; Taylor, J. Amplification and direct sequencing of fungal ribosomal RNA genes for phylogenetics. In PCR Protocols: A Guide to Methods and Applications; Innis, N., Gelfand, D., Sninsky, J., White, T., Eds.; Academic Press: Cambridge,MA, USA, 1990; pp. 315–322.
Line 159: UNITE database. The authors should cite which version of the database they used. If they used the online version, then they should add the date they assessed the online version.
Lines 178-179: If these formulas are standard formulas, they are unnecessary and should be removed.
Line 229: preferred is an opinion and not a result. Please add a more neutral word like “ Sordariomycetes (1.93-50.42%) were more abundant in the summer, while Dothideomycetes (11.2-72.46%) were more abundant in the winter”.
All charts: For some of the charts were required, the authors added the Abbreviations of SM, BC, AM, AL, LJ but not for all. Please make sure the definitions for all the abbreviations are included in the Figure legends.
Figure 2: Line 256: adjust the spacing after the abbreviations so they are all the same.
Figure 3 and Figure 4: Suggest changing OUT to OTU, occurs on lines 266-267, 274-275.
Figure 5: Seems to lack a stand-a-lone title. Please add a title and then use (a) Alpha diversity, (b) Niche width.
Line 302: I believe the authors have checked “The SPSS 26.0 software was employed to check the normal distribution and variance homogeneity of the experimental data. The data for the OTU richness of fungi were consistent with the normal distribution and homogeneity of variance,” however, it was not clear that the datasets for the PERMANOVAs were tested for equal variance for all the groupings tested using the final datasheet. There should be a statistical test for each grouping to verify that the variance within each grouping was not statistically different in each set of groupings. If it was, then it breaks the underlying assumption of the test. Please clarify.
Table 1: The * should be explained in a foot note.
Line 351: Typically, first words of a sentence should be spelled out. Please adjust or add: The MCODE analysis…
Line 360 and 362: Results should be written in past tense: please adjust shows to showed, and changes to changed.
Table S2: As previously mentioned, this was meant to be Table S1.
3.5 line 395: In this section the authors must add the percentage of OTUs that were assigned to a function prediction and the percentage of those not assigned.
Line 445: suggest changing did to do
Data availability, line 623: The authors should also add: Raw sequence data was submitted to the NCBI Sequence Read Archive (SRA) project accession number PRJNA942110.
Comments on the Quality of English LanguageNA
Author Response
Thank you very much for your valuable comments and advice. In the revision, we carefully considered the editor’s and reviewers’ comments and suggestions. The actions taken in the revision and answers and explanations in response to the queries are listed below.
Responses to the reviewers’ comments:
Reviewer 4
Comment 1: The manuscript is well written and concise overall. The manuscript is scientifically sound for the most part but at least one clarification is needed.Clarification needed: The authors use FUNGuild for functional evaluation; however, the authors need to state how many of the OTUs were functionally characterized and how many were not characterized. This is important because the authors discuss increased pathogens or increased saprotrophs, however it is important for the reader to know if this is based on 100% of the OTUs assigned to a guild or only 50% of the OTUs assigned to a guild. Typically, with NGS data a lot of the OTUs are unassigned so this needs to be added.
Response: Thanks for your comment. We have made the supplement according to your requirements. (Please see Line422-424 in the revised manuscript).
Comment 2: Line 110: This is not the correct table. As per the manuscript, Table S1 contains the keystone species and Table 2 shows the climate parameters. Please adjust.
Response: Thanks for your comment and suggestions. We have already adjusted the order of the TableS1 and TableS2. (Please see Line 132,417 in the revised manuscript).
Comment 3: Line 120: This is awkward as written: suggestion: In total, 120 samples were processed for analyses.
Response: Thanks for your comment. We have revised. (Please see Line129-131in the revised manuscript).
Comment 4: Current table S1 (Keystone species). Unclassified Glomeromycota is not an Order so the column heading should be adjusted. This appears in two separate locations. Also, it is unclear where the total starts so a line separating the individual data vs the totals would visually make it clearer.
Response: Thanks for your comment. We have revised. (Please see TableS2 in the revised manuscript).
Comment 5: Line 140: The ITS2R primer is identical to the primer developed by White et al. 1990 so the authors should cite the original designer of the primer.
White, T.J.; Bruns, T.; Lee, S.; Taylor, J. Amplification and direct sequencing of fungal ribosomal RNA genes for phylogenetics. In PCR Protocols: A Guide to Methods and Applications; Innis, N., Gelfand, D., Sninsky, J., White, T., Eds.; Academic Press: Cambridge,MA, USA, 1990; pp. 315–322.
Response: Thanks for your comment. We have revised. (Please see Line145,868-870 in the revised manuscript).
Comment 6:Line 159: UNITE database. The authors should cite which version of the database they used. If they used the online version, then they should add the date they assessed the online version.
Response: Thanks for your comment and suggestions. We have revised. (Please see Line165-167in the revised manuscript).
Comment 7:Lines 178-179: If these formulas are standard formulas, they are unnecessary and should be removed.
Response: Thanks for your comment. Other reviewers suggested keeping and improving the relevant formulas. We have supplemented the formulas involved. (Please see Line190-192in the revised manuscript).
Comment 8:Line 229: preferred is an opinion and not a result. Please add a more neutral word like “ Sordariomycetes (1.93-50.42%) were more abundant in the summer, while Dothideomycetes (11.2-72.46%) were more abundant in the winter”.
Response: Thanks for your comment and suggestions. We have removed the relevant formulas. (Please see Line 242-244 in the revised manuscript).
Comment 9:All charts: For some of the charts were required, the authors added the Abbreviations of SM, BC, AM, AL, LJ but not for all. Please make sure the definitions for all the abbreviations are included in the Figure legends.
Response: Thanks for your comment. We have revised.(Please see Line 267-269, 279-281, 288-291, 310-312,333-335,360-362,398-400 and 450-453 in the revised manuscript).
Comment 10:Figure 2: Line 256: adjust the spacing after the abbreviations so they are all the same.
Response: Thanks for your comment. We have revised. (Please see Line 267-269 in the revised manuscript).
Comment 11:Figure 3 and Figure 4: Suggest changing OUT to OTU, occurs on lines 266-267, 274-275.
Response: Thanks for your comment. We have revised. (Please see Line 277-278,287-288 in the revised manuscript).
Comment 12:Figure 5: Seems to lack a stand-a-lone title. Please add a title and then use (a) Alpha diversity, (b) Niche width.
Response: Thanks for your comment and suggestions. We have revised. (Please see Line 297-298 in the revised manuscript).
Comment 13:Line 302: I believe the authors have checked “The SPSS 26.0 software was employed to check the normal distribution and variance homogeneity of the experimental data. The data for the OTU richness of fungi were consistent with the normal distribution and homogeneity of variance,” however, it was not clear that the datasets for the PERMANOVAs were tested for equal variance for all the groupings tested using the final datasheet. There should be a statistical test for each grouping to verify that the variance within each grouping was not statistically different in each set of groupings. If it was, then it breaks the underlying assumption of the test. Please clarify.
Response: Thanks for your comment. We have already supplemented the content using variance test. (Please see Line 320-323 in the revised manuscript).
Comment 14:Table 1: The * should be explained in a foot note.
Response: Thanks for your comment. We have revised. (Please see Line 326 in the revised manuscript).
Comment 15:Line 351: Typically, first words of a sentence should be spelled out. Please adjust or add: The MCODE analysis…
Response: Thanks for your comment. We have revised.(Please see Line 373 in the revised manuscript).
Comment 16:Line 360 and 362: Results should be written in past tense: please adjust shows to showed, and changes to changed.
Response: Thanks for your comment and suggestions. We have revised. (Please see Line 381,383 in the revised manuscript).
Comment 17:Table S2: As previously mentioned, this was meant to be Table S1.
Response: Thanks for your comment and suggestions. We have made the modifications to this part. (Please see Line 417 in the revised manuscript).
Comment 18:3.5 line 395: In this section the authors must add the percentage of OTUs that were assigned to a function prediction and the percentage of those not assigned.
Response: Thanks for your comment. We have made the supplement according to your requirements. (Please see Line422-427 in the revised manuscript).
Comment 19:Line 445: suggest changing did to do.
Response: Thanks for your comment. We have revised. (Please see Line 478 in the revised manuscript).
Comment 20:Data availability, line 623: The authors should also add: Raw sequence data was submitted to the NCBI Sequence Read Archive (SRA) project accession number PRJNA942110.
Response: Thanks for your comment. We have revised. (Please see Line 658-659 in the revised manuscript).
Round 2
Reviewer 1 Report
Comments and Suggestions for Authors
The manuscript is clear; the authors have made substantial revisions and addressed all of the concerns from a reviewed previous version.
I agree with the modifications made by the authors in response to all the comments. But minor corrections must be attended to the manuscript before being accepted.
L118: Instead of the year, please specify the age of the plants at the moment of sampling. This is relevant because the physiological state in plants determines their microbiota, as has been demonstrated in: https://doi.org/10.1016/j.soilbio.2021.108468
https://doi.org/10.1186/s40793-022-00415-3
https://doi.org/10.1128/spectrum.01834-21
L346: please correct “Te fgure” to “The figure”
L515-516: Please, correct the species and genera names using the italics letter format.
The authors improved substantially the figures' quality. However, the figures' legends are still blurry and small, particularly the Fig. 5 axes legends of the graphic and the right column series legends in Figures 1, 5, 8, and 11.
Author Response
Thank you very much for your valuable comments and advice. In the revision, we carefully considered the editor’s and reviewers’ comments and suggestions. The actions taken in the revision and answers and explanations in response to the queries are listed below.
Responses to the reviewers’ comments:
Reviewer 1
Comment 1:L118: Instead of the year, please specify the age of the plants at the moment of sampling. This is relevant because the physiological state in plants determines their microbiota, as has been demonstrated in: https://doi.org/10.1016/j.soilbio.2021.108468
https://doi.org/10.1186/s40793-022-00415-3
https://doi.org/10.1128/spectrum.01834-21
Response:Thanks for your comment.We have revised. (Please see Line 121-122 in the revised manuscript).
Comment 2:L346: please correct “Te fgure” to “The figure”
Response: Thanks for your comment.We have revised. (Please see Line 353 in the revised manuscript).
Comment 3:L515-516: Please, correct the species and genera names using the italics letter format.
Response: Thanks for your comment and suggestions.We have revised. (Please see Line 529-530 in the revised manuscript).
Comment 4:The authors improved substantially the figures' quality. However, the figures' legends are still blurry and small, particularly the Fig. 5 axes legends of the graphic and the right column series legends in Figures 1, 5, 8, and 11.
Response: Thanks for your comment.We have revised.(Please see Fig.1,5,8 and11 in the revised manuscript).
Reviewer 4 Report
Comments and Suggestions for Authors
Overall, the authors have addressed my concerns. However, there are some minor corrections throughout the paper.
Methods and materials: should be written in the past tense.
Line 120: are should be changed to were
Line 121: is should be changed to was
Line 122: Please check to see if there is a space between individual. Fresh
Line 168: suggest changing using an RDP classification to using the RDP classification
Line 169-170: This sentence should be removed as there is no mention of more than one dataset above. Appears out of place and if the authors completed this, they should mention both, and then state this.
Line 248: suggest changing were found in leaf to--were found in the leaves.
Line284-285: All of the "of" should be changed to "at". Also, same for figure 4 lines 294-295.
Comments on the Quality of English LanguageThere are a few grammatical errors throughout.
Author Response
Thank you very much for your valuable comments and advice. In the revision, we carefully considered the editor’s and reviewers’ comments and suggestions. The actions taken in the revision and answers and explanations in response to the queries are listed below.
Responses to the reviewers’ comments:
Reviewer 4
Comment 1: Line 120: are should be changed to were.
Response: Thanks for your comment. We have revised.(Please see Line 124 in the revised manuscript).
Comment 2: Line 121:is should be changed to was.
Response: Thanks for your comment and suggestions. We have revised. (Please see Line 124 in the revised manuscript).
Comment 3: Line 122: Please check to see if there is a space between individual. Fresh.
Response: Thanks for your comment. We have revised. (Please see Line125 in the revised manuscript).
Comment 4: Line 168: suggest changing using an RDP classification to using the RDP classification.
Response: Thanks for your comment and suggestions.We have revised. (Please see Line172 in the revised manuscript).
Comment 5: Line 169-170: This sentence should be removed as there is no mention of more than one dataset above. Appears out of place and if the authors completed this, they should mention both, and then state this..
Response: Thanks for your comment. We have already deleted this sentence.
Comment 6:Line 248: suggest changing were found in leaf to--were found in the leaves.
Response: Thanks for your comment. We have revised. (Please see Line 253 in the revised manuscript).
Comment 7:Line284-285: All of the "of" should be changed to "at". Also, same for figure 4 lines 294-295.
Response: Thanks for your comment.We have revised. (Please see Line289-290,300-301in the revised manuscript).